# Recombinant human adenovirus p53 combined with transcatheter arterial chemoembolization for liver cancer: A meta-analysis

**Yaru Guo**[1☯], **Yuanyuan Chen**[1☯], **Yingnan Zhang**[1], **Mengjun Xu**[1], **Wenwen Guo**[1], **Jingya Zhang**[1], **Gaolei Ma**[1], **Chen Liu**[2], **Juan Yang**[2], **Xiaojin Wu**[1]*

1 Department of Radiation, Xuzhou first People's Hospital, Jiangsu, China, 2 Xuzhou Medical University, Jiangsu, China

☯ These authors contributed equally to this work.
* 1047821144@qq.com

**Data Availability Statement:** All relevant data are within the paper and its Supporting Information files.

## Abstract

### Objectives

To compare the clinical curative effects, survival and complications of recombinant human adenovirus-p53 (rAd-p53) combined with transcatheter arterial chemoembolization (TACE) versus TACE for the treatment of liver cancer.

### Methods

We searched all the eligible studies of rAd-p53 plus TACE versus control group had only TACE in the treatment of liver cancer, which were retrieved from CNKI, Wanfang database, CBM, VIP, PubMed, EMBase, The Chrance of Library, Web of Science from its inception to august 2022.

### Results

A total of 17 studies were included, which involved 1045 patients. The results of the meta analysis indicated that the the rAd-p53combined with TACE markedly improved the patients' complete remission(OR = 2.19, 95% CI:1.13–4.22, P = 0.02), partial remission (OR = 2.22, 95% CI:1.67–2.94, P<0.00001), objective tumor response rate (OR = 2.58, 95% CI:1.95–3.41, P<0.00001) and disease control rate(OR = 2.39, 95% CI:1.65–3.47, P<0.00001) compared with TACE alone. And our results showed that rAd-p53combined with TACE had better survival benefit [6-month OS (OR = 3.41, 95% CI: 1.62–7.14, p = 0.001); 1-year OS (OR = 1.95, 95% CI: 1.28–2.96, p = 0.002)] and better quality of life(MD = 5.84, 95% CI:2.09–9.60, P = 0.002). In addition, the immunity of the patients was enhanced by the combination therapy, as demonstrated by the increase in the ratio of CD4+ to CD4+/CD8+. In adverse effects, except for fever in the TACE combined with rAd-p53 group, which was higher than that in the TACE group(OR = 2.62, 95% CI:2.02–3.49, P<0.00001), all other adverse effects were lower in the TACE combined with rAd-p53 group than in the TACE group.

**Funding:** This work was supported by the clinical medicine expert team project of Academician Wu Mengchao of the Eastern Hepatobiliary Surgery Hospital affiliated to the Naval Military Medical University, introduced to the Xuzhou First People's Hospital.This work was also supported by Wu Jieping Foundation, Xuzhou Health System Reserve Talent Fund, Jiangsu Province Medical Young Professionals Fund and Jiangsu Provincial Committee Organization Department 333 Talent Fund. The funders played an important role in guiding our study design and data analysis.The funders, who has been involved in research on liver cancer for many years, played an important role in guiding our project design and provided financial support for our project design, collection efforts and final statistical analysis.

**Competing interests:** The authors have declared that no competing interests exist.

## Conclusion

RAd-p53 combined with TACE for liver cancer showed significant advantages in terms of clinical efficacy, survival rate, and safety compared to the TACE alone, and effectively improved patient quality of life and immune function.

## Systematic review registration

https://inplasy.com/inplasy-2022-9-0127/.

## Introduction

Primary liver cancer (PLC) is a malignant tumor that commonly originates from epithelial cells of hepatocytes or intrahepatic bile ducts. According to the International Agency for Research on Cancer (IARC) 2020, primary liver cancer is considered the sixth most common cancer and the third leading cause of cancer death worldwide [1], with more than 905,677 new cases and 830,180 new deaths recorded globally in recent years [2].

Surgical treatment, including hepatectomy and liver transplantation, has long been the backbone of hepatocellular carcinoma treatment, surgery requires consideration of the patient's liver function, the presence and extent of portal hypertension, performance status, and tumor characteristics, such as the size, number, and extent of involvement of the liver and portal veins [3,4]. However, most patients are already in the middle or advanced stages of liver cancer at the time of diagnosis, so only about 20–30% of patients have the opportunity to undergo hepatectomy [5]. Moreover, the recurrence rate of liver cancer after surgery is as high as 70–80 percent. [6]. Although liver transplantation is feasible, it is difficult to meet clinical needs due to the insufficient number of donors [7]. In the last few years, imaging-guided ablative therapies have gained a fundamental role in the treatment of HCC, due to their safety and efficacy, leading to complete necrosis of the tumoral nodule. Among them, percutaneous radiofrequency ablation (RFA) has become the standard-of-care for unresectable early HCCs and has been even found to be competitive with surgery in the case of a single nodule less than 3 cm in size [8]. However, the significant incidence of local and distant recurrences was found to affect survival and the procedure can be complex and risky when the tumor is adjacent to the gallbladder, major blood vessels or diaphragm [9].Transcatheter arterial chemoembolization (TACE), has become has become the most common treatment for unresectable hepatocellular carcinoma [10].TACE focuses chemotherapy drugs on the HCC site while blocking the tumor blood supply artery, and can improve the survival rate of tumor patients in the middle and late stages [11]. The most common drugs used during conventional TACE are doxorubicin, epirubicin or cisplatin [12]. Although TACE was effective in slowing the progression of HCC early after treatment, it was less effective when used for longer than 6 months and had a 2-year survival rate of 24–63% [13].Therefore, it is necessary to combine gene therapy or radiotherapy with TACE to improve patient outcomes.

The tumor suppressor gene p53 is widely recognized as the genetic guardian of cells and plays a key role in cell cycle control, apoptosis and inhibition of tumor cell proliferation and more than 60% of human tumor lesions are associated with mutations in the p53 gene [14]. Recombinant human p53 adenovirus (rAd-p53) is the first gene therapy drug approved for clinical use produced by Shenzhen SiBiono GeneTech [15]. RAd-p53 can transfer the p53 gene into tumor cells via adenovirus to reconstruct the p53 gene function of tumor cells and cause

programmed death or severe dormancy of tumor cells, thus improving the anti-tumor efficacy [16].Studies have shown that recombinant human p53 adenovirus can effectively enhance the immune function of the body, and the combination with TACE in the treatment of primary liver cancer can make up for the shortcomings of monotherapy [17].

This study aimed to systematically evaluate the clinical curative effect, survival rate and complications of rAd-p53 combined with TACE for hepatocellular carcinoma through a meta-analysis.

## Materials and methods

INPLASY registration number: This protocol was registered with the International Platform of Registered Systematic Review and Meta-Analysis Protocols (INPLASY) on 30 September 2022 and was last updated on 30 September 2022 (registration number INPLASY202290127).

### Study inclusion criteria

i. Clinical trials of rAd-p53 combined with TACE in the treatment of liver cancer.

ii. Patients diagnosed with liver cancer by pathology or clinical examination and did not receive surgery or other treatment before participating in the clinical trial.

iii. The experimental group received rAd-p53 combined with TACE, while the control group received TACE alone.

iv. Outcome indicators: complete remission(CR), partial remission (PR), objective tumor response rate (ORR), disease control rate(DCR) which was according to the internationally accepted criteria for evaluating the efficacy of solid tumors (RECIST 1.1). and overall survival (OS) of 6 months, 1 or 2 years were used as primary effificacy outcomes. Also includes the Quality of life(QoL) which was evaluaded using he Karnofsky score (KPS), All of the included literature evaluated patients' quality of life according to the Karnofsky scale: patients were divided into ten levels according to their condition and the degree of normal activity and self-care, with 10 points/level and a total score of 100, and patients' health status and quality of life were positively correlated with the score. The change of AFP, immune indexes (CD3+, CD4+, and CD8+ percentages and CD4+/CD8+ cell ratios) and adverse effects.

### Study exclusion criteria

i. Non-interventional clinical trials or observational or retrospective studies.

ii. The studies which enrolled patients had had other treatments before enrollment. For studies with inconsistent intervention measures, for example, except for rAd-p53, there are other inconsistent treatment measures between the experimental group and the control group. In addition, single arm studies, studies with inconsistent design and outcome indicators should be excluded.

iii. Review, animal or basic experiment, case report and literature irrelevant to research content.

## Search strategy and study selection

We comprehensively searched for clinical studies of rAd-p53 combined with TACE for liver carcinoma in the PubMed, the Cochrane of Library, Web of Science, Embase, Wanfang Data, Chinese National Knowledge Infrastructure (CNKI), Chinese Biological Medicine (CBM) Database, and VIP Database until August 2022.We retrieved the following terms from the Medical Subject Headings (MeSH): Liver Neoplasms; Neoplasms, Hepatic; hepatocellular carcinoma; Recombinant human adenovirus-p53; rAd-p53; transcatheter arterial chemoembolization. Two researchers(YG and YZ) independently screened the literature in strict accordance with the inclusion and exclusion criteria. In case of disagreement, the third researcher decides whether toinclude the study.

## Data extraction and quality assessment

Relevant data were extracted independently by two authors (YG and MX) according to predesigned data tables, including authors, year of publication, country, sample size, age, sex ratio, interventions, rAd-p53 dose, TACE regimen, CR/PR/ORR/DCR rates, 6-month, 1,2 year-OS, Changes in AFP, KPS score and immune indexes and complications after treatment in the experimental and control groups. Two independent authors strictly assessed the quality of RCT studies according to the tools provided by the Cochrane Collaboration Network, using MINORS entries to evaluate the quality of non-randomized clinical trials.MINORS scores 0–8 literature as low quality, 9–16 as moderate quality and 17–24 as high quality.

## Statistical analysis

Review Manager (RevMan) version 5.3.5 and Stata SE12.0 (Stata Corporation, College Station, Texas, USA) were used to perform a meta-analysis of the data from included studies. The outcome results of dichotomous data will be analyzed as odds ratios (ORs) and continuous data will be analyzed as mean difference (MD) or standardized MD, 95% confidence intervals (CIs) being reported for all estimates. Heterogeneity between studies was measured using the Q statistic (Chi-square test) and the Higgins $I^2$ index. When $P > 0.1$ and $I^2 < 50.0\%$, heterogeneity was considered insignificant and a fixed-effects model was used. When $P < 0.1$ and $I2 > 50.0\%$ indicated significant heterogeneity, a fixed-effects model was used, along with subgroup analysis or sensitivity analysis or meta-regression to identify potential causes of heterogeneity. The results of this meta-analysis were presented in the form of a forest plot. Egger's and Begg's tests were used to assess potential publication bias, and Egger's and Begg's funnel plots were also drawn.If significant publication bias was detected, the non-parametric "trim and fill" method was used to adjust for the bias. All tests were two-sided and P values less than 0.05 were considered statistically significant.

# Results

## Characteristics of studies

We initially retrieved a total of 864 publications, excluding 286 duplicates, 368 studies with irrelevant content, 106 reviews and case reports, 50 animal experiments or basic experiments. After reading through the full text, we excluded 37 studies with inconsistent experimental design, included 3 retrospective studies, 13 single-arm studies, 6 studies with inconsistent outcome indicators, and 15 studies with inconsistent interventions. A total of 17 studies were finally included for meta-analysis, and the literature screening process is shown in **Fig 1**. 1045 patients were included, 491 in the experimental group and 554 in the control group. Eight of these were based on randomized controlled clinical trials, and the other nine were non-

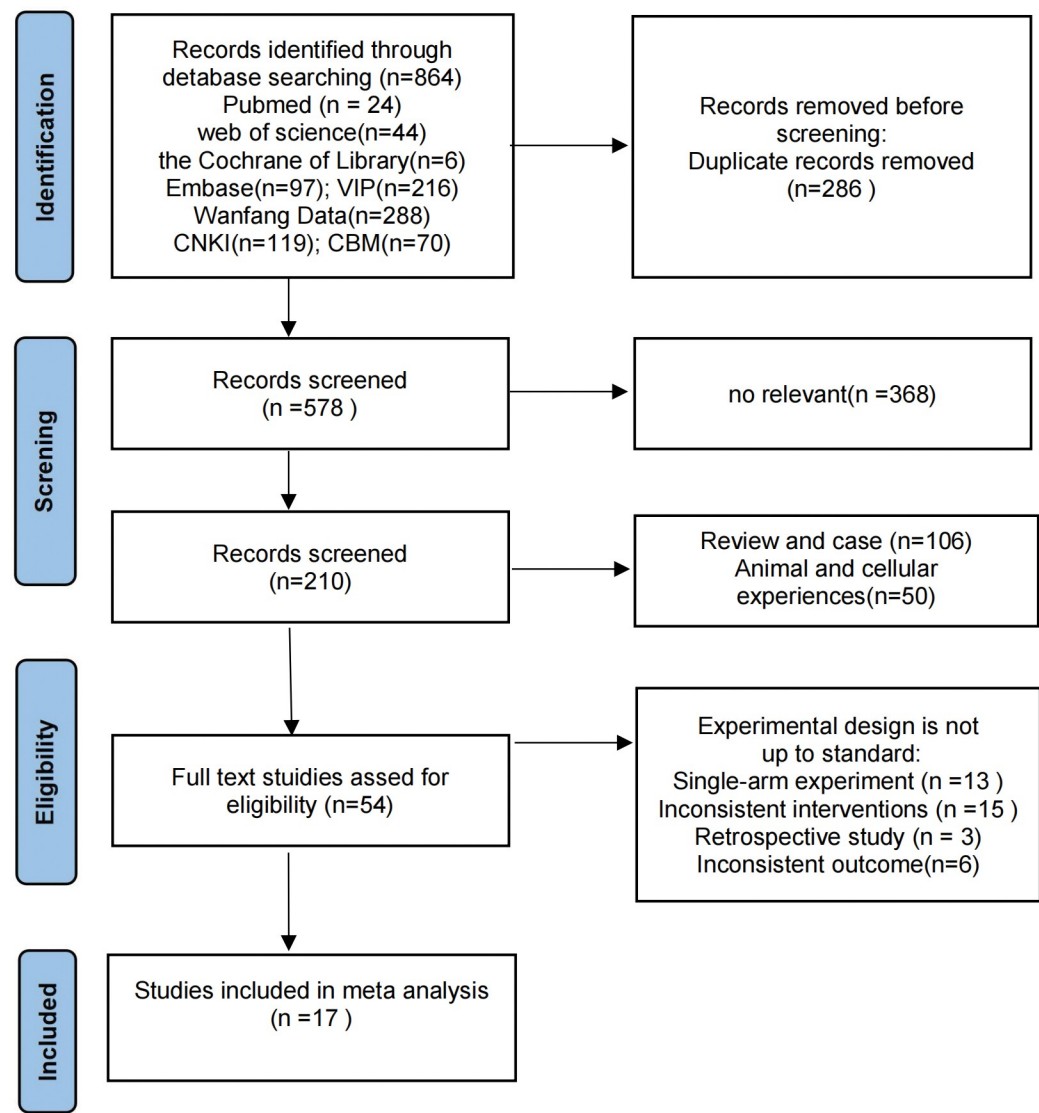

**Fig 1. Flow chart of studies screening.**

randomized controlled clinical trials. Patients in the experimental group received rAd-p53 combined with TACE treatment, while patients in the control group received only TACE treatment.Under local anesthesia, a 5 F hepatic tube was punctured through the femoral artery and placed in the common hepatic artery and superior mesenteric artery for imaging to clarify the number, location, size, blood supply vessels and the presence of arteriovenous fistula of the tumor. Then, microcatheters were used to superselect to the tumor target vessels, and epirubicin, cisplatin. Before the injection of chemotherapy drugs, rAd-p53 was diluted to 10–15 mL with saline and injected slowly into the tumor artery through the catheter. rAd-p53 dosage was determined according to the patients' physical condition, tumor size and tumor number.All rAd-p53 used in the included studies was produced by Shenzhen Sebenor Gene Technology Co, Ltd with a specification of 1×1012VP/stem. The study and patient characteristics are summarized in **Table 1**.

**Table 1. Characteristics of studies included.**

| study | Study design | Sample size (Exp/Con) | year | | Gender Male/female | | Interventions | | rAd-p53 dose | TACE regimens | MINORS |
|---|---|---|---|---|---|---|---|---|---|---|---|
| | | | Exp | Con | Exp | Con | Exp | Con | | | |
| Zhu CQ 2017[24] | No-RCT | 30/30 | 58.21 ±5.49 | 58.43±5.21 | 15/15 | 17/13 | rAd-p53+ TACE | TACE | $1–4\times10^{12}$VP time /4weeks/2cycles | Epirubicin:6mg/m2 Cisplatin:60mg/m2 5-Fu:2000mg | 16 |
| Tang YM 2016[19] | No-RCT | 20/20 | 43–76 | 48–77 | 11/9 | 14/6 | rAd-p53+ TACE | TACE | $2–3\times10^{12}$VP time/ 4weeks/4-5cycles | Epirubicin:6mg/m2 Cisplatin:35-55mg/m2 5-Fu:2000mg | 17 |
| Ai Shen 2015[23] | RCT | 23/25 | 49.1± 13.1 | 48. 6±11. 2 | 17/6 | 18/7 | rAd-p53+ TACE | TACE | $2–3\times1012$VP time/ 4weeks/4-5cycles | Epirubicin:25-35mg/m2 Cisplatin:60mg/m2 5-Fu:2000mg | - |
| Cui HL 2015 [20] | No-RCT | 49/53 | 53.9± 10.9 | 53.2 ± 12.1 | 39/10 | 41/12 | rAd-p53+ TACE | TACE | - | - | 18 |
| Li DH 2015 [29] | No-RCT | 36/36 | 52.5 ±10.8 | 52.5±10.8 | - | - | rAd-p53+ TACE | TACE | $1\times10^{12}$VP time 4weeks/4cycles | Pharmorubicin:5mg/m2 Mitomycin:5-10mg/m2 Oxaliplatin:130mg/m2 | 17 |
| Shixi Chen 2014[30] | RCT | 29/32 | 54.3(47–65) | 54.3(47–65) | 21/8 | 23/9 | rAd-p53+ TACE | TACE | $1–3\times10^{12}$VP time /4weeks/3cycles | Pharmorubicin Cisplatin 5-Fu | - |
| Yang LQ 2013[25] | RCT | 30/18 | 53(34–76) | 51(28–68) | 25/5 | 14/4 | rAd-p53+ TACE | TACE | $1\times10^{12}$VP Time /weeks/3cycles | Camptothecin:20mg/m2 | |
| Jiang M 2012[21] | No-RCT | 23/25 | 43.7(39–75) | 43.7(39–75) | - | - | rAd-p53+ TACE | TACE | $1\times10^{12}$VP time /4weeks/2-4cycles | Navelbine Camptothecin 5-Fu | 19 |
| Hou LN 2012[34] | RCT | 18/18 | (42.8 ±3.6) | (43.9±3.9 | 13/5 | 14/4 | rAd-p53+ TACE | TACE | $2\times10^{12}$VP time 4weeks/1 cycles | Hydroxycamptothecin Adriamycin 5-Fu | - |
| Liang AJ 2011[31] | RCT | 30/30 | - | - | - | - | rAd-p53+ TACE | TACE | $2\times10^{12}$VP time /2weeks/3cycles | Cisplatin:60-80mg/m2 | - |
| Xu WG 2011 [26] | RCT | 26/26 | 48.3(32–66) | 49.0(34–72) | 25/1 | 26/0 | rAd-p53+ TACE | TACE | - | - | - |
| Guo ZJ 2010 [22] | No-RCT | 21/32 | 52.6 ±10.9 | 50.5±9.8 | 17/4 | 25/7 | rAd-p53+ TACE | TACE | $1\times10^{12}$VP time 4weeks/1 cycles | Mitomycin:10-20mg Pharmorubicin:30-80mg Oxaliplatin: 150-250mg | 20 |
| Ou SQ 2010[27] | RCT | 20/20 | 36.5(25–70) | 36.5(25–70) | 15/5 | 14/6 | rAd-p53+ TACE | TACE | $3\times10^{12}$VP time 4weeks/3cycles | Epirubicin:40-60mg Oxaliplatin:50-100mg | - |
| Peng XB 2009[28] | No-RCT | 60/60 | 19–71 | 18–74 | 45/15 | 41/19 | rAd-p53+ TACE | TACE | $2\times10^{12}$VP time 4weeks/3-4cycles | Hydroxycamptothecin:30mg Adriamycin: 50mg 5-Fu: 1000mg | 17 |
| Mo YX 2009 [32] | No-RCT | 22/56 | 31–74 | 32–76 | 18/4 | 47/9 | rAd-p53+ TACE | TACE | $1\times10^{12}$VP time 4weeks/3cycles | Mitomycin:10-20mg Pharmorubicin:50-100mg Carboplatin:300-700mg | 16 |
| Zhu ZB 2005 [33] | RCT | 28/34 | 48.25 ±11.25 | 49.15 ±10.76 | 19/9 | 24/10 | rAd-p53+ TACE | TACE | $2–4\times10^{12}$VP time 4weeks/2cycles | Pirarubicin:30mg/Mitomycin 8mg.Oxaliplatin:50-100mg/ cisplatin:40-80mg 5-Fu:500-1090mg | - |
| Zhu CQ 2017[24] | No-RCT | 26/39 | - | - | - | - | rAd-p53+ TACE | TACE | $2\times10^{12}$VP time 4weeks/4cycles | - | 18 |

## Quality assessment

The results of the quality evaluation of the eight RCT studies are shown in **Fig 2**. Four of the studies assigned random numbers, while none of the remaining studies described any particular method of randomization. four studies included patients who signed informed consent and were explicitly not blinded, and the remaining four studies did not have sufficient information

**A**

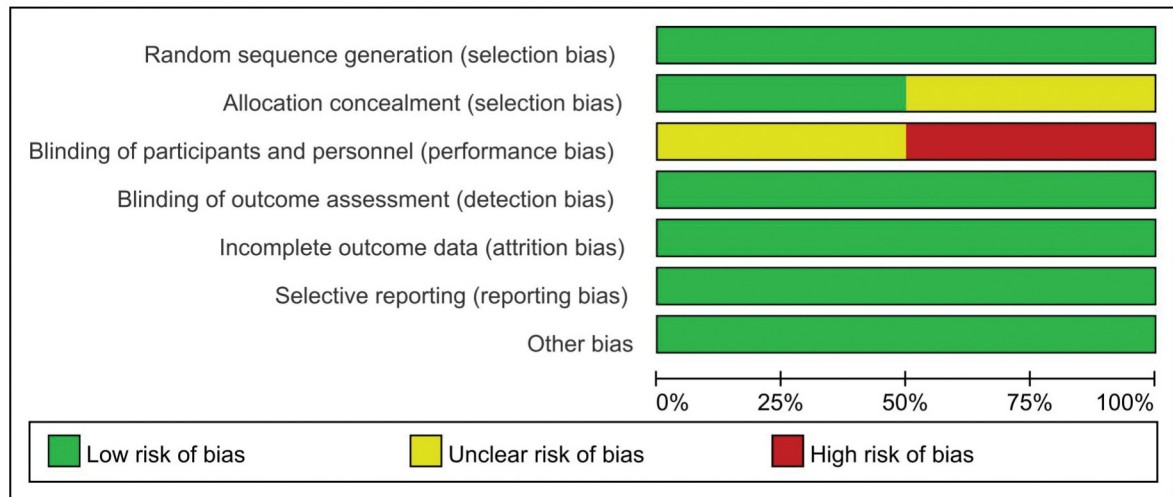

**B**

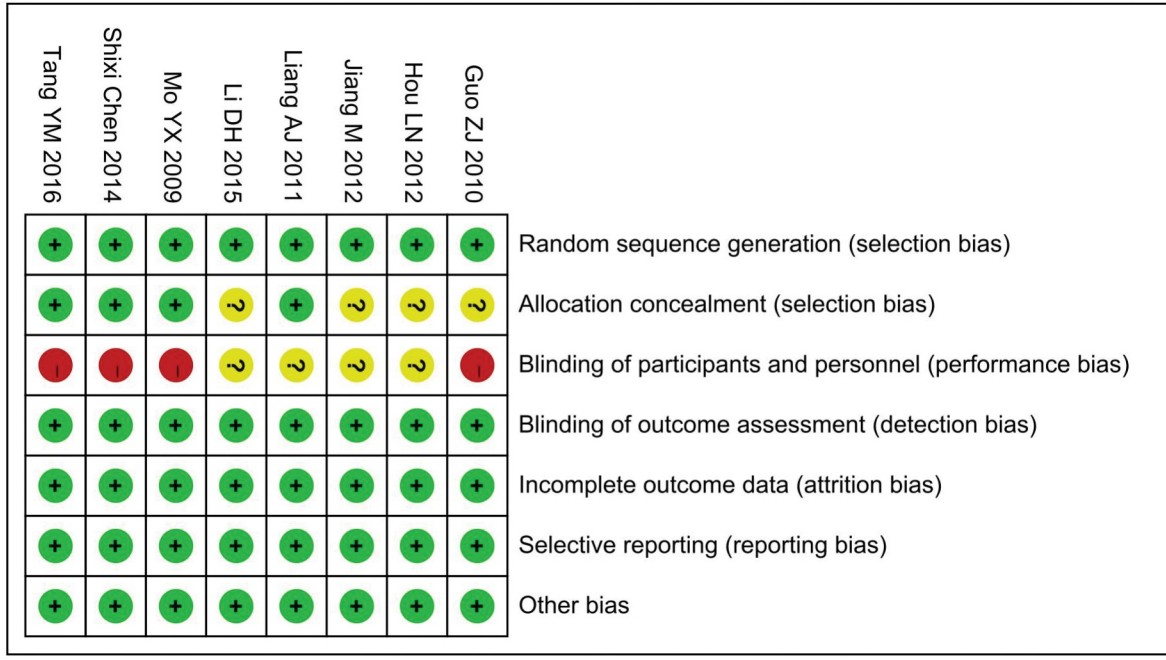

**Fig 2.** Risk of bias summary (A) and bias graph (B). Green indicates low risk; Red indicates high risk; Yellow indicates unknown risk.

to determine whether blinding was used. All included studies submitted complete data for analysis, and no selective reporting or other bias was found. Minors was used to assess the quality of the nine non-randomized clinical studies, and the results showed that all scores were greater than or equal to 16, as assessed in **Table 1**. Quality analysis showed that all the included studies were of high or moderate quality.

## Efficiency

**CR.** Fix studies [18–22] reported CR rates. There was nosignificant heterogeneity between studies (p = 0.68, $I^2$ = 0%); therefore, the fixed-effects model was used for the meta-analysis.

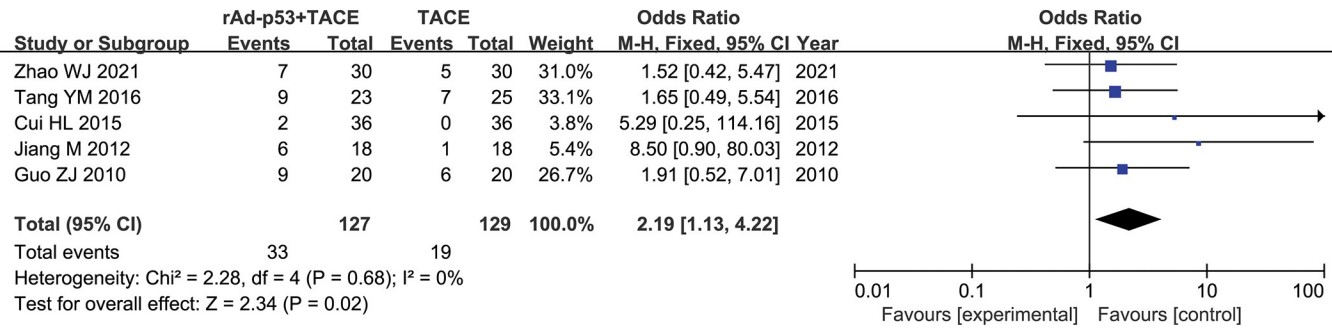

**Fig 3. Forest plot for CR of rAd-p53 combined with TACE group and TACE alone group.**

the results of the Meta-analysis showed that the rAd-p53 combined with TACE group had better CR rates than the TACE group (OR = 2.19, 95% CI:1.13–4.22, P = 0.02) (**Fig 3**).

**PR.**   Fifteen studies[18–32] reported PR rates. There was nosignificant heterogeneity between studies (p = 0.97, $I^2$ = 0%); therefore, the fixed-effects model was used for the meta-analysis. the results of the Meta-analysis showed that the rAd-p53 combined with TACE group had better CR rates than the TACE group (OR = 2.22, 95% CI:1.67–2.94, P<0.00001) (**Fig 4**).

**ORR.**   Sixteen studies [18–33] reported ORR rates. There was nosignificant heterogeneity between studies (p = 0.92, $I^2$ = 0%); therefore, the fixed-effects model was used for the meta-analysis. The results of the meta-analysis showed a significantly higher ORR rate in the rAd-p53 combination TACE compared to that in the TACE alone group (OR = 2.58, 95% CI:1.95–3.41, P<0.00001) (**Fig 5**).

**DCR.**   Fifteen studies [18–32] reported DCR rates. There was nosignificant heterogeneity between studies (p = 0.96, $I^2$ = 0%); therefore, the fixed-effects model was used for the meta-analysis. The results of the meta-analysis showed that the ORR rate were higher in the rAd-p53 combination TACE than TACE alone group (OR = 2.39, 95% CI:1.65–3.47, P<0.00001) (**Fig 6**).

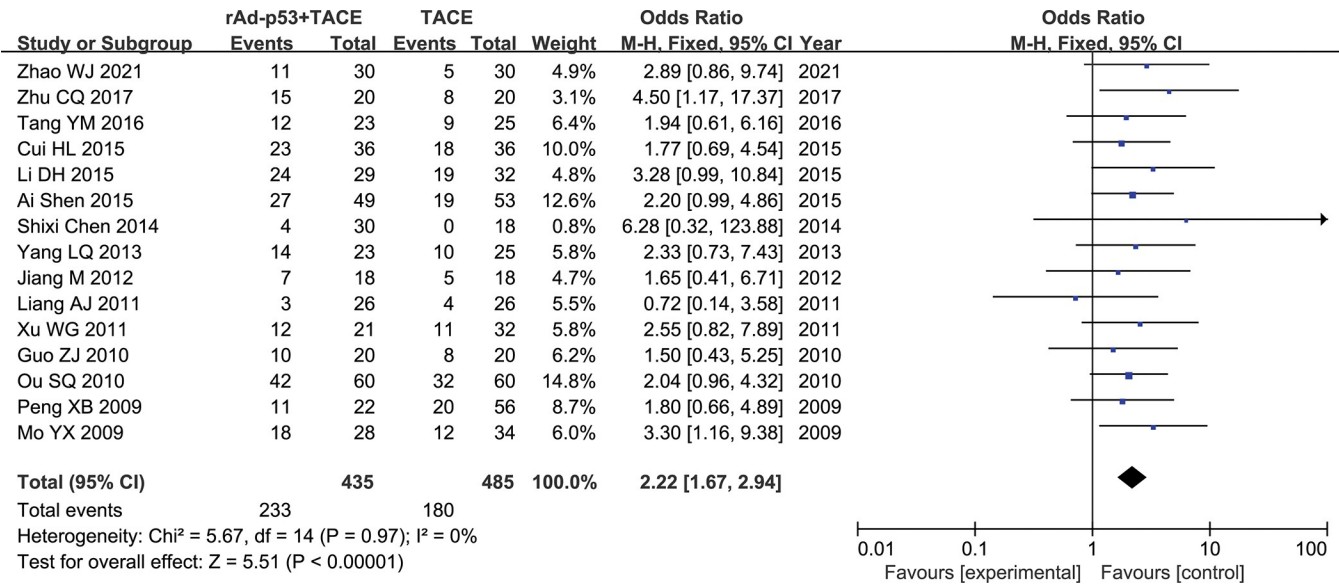

**Fig 4. Forest plot for PR of rAd-p53 combined with TACE group and TACE alone group.**

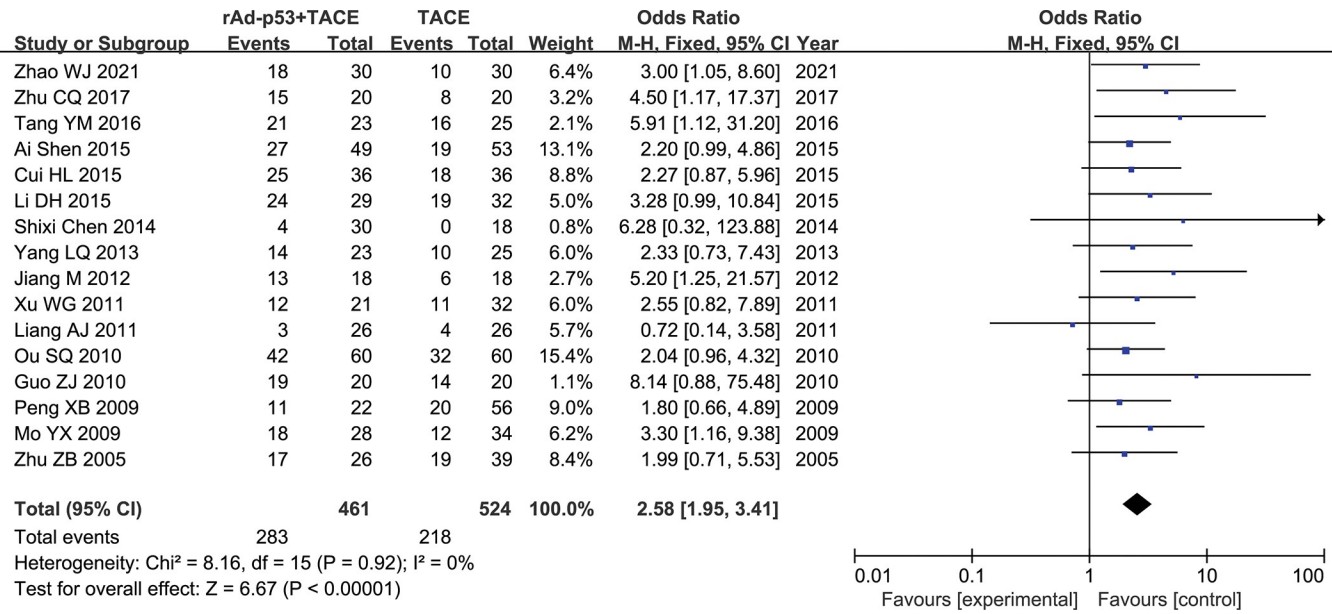

**Fig 5. Forest plot for ORR of rAd-p53 combined with TACE group and TACE alone group.**

## 6-month, 1-year OS and 2-year OS

Four studies [22,25,26,31] reported six-month survival rates; nine studies [18,22–24,26,29–31,34] reported one-year survival rates; three studies [23,26,31] reported two-year survival rates. There was no heterogeneity among the studies, so a fixed-effects model was used(6 month-OS: p = 0.19, I2 = 37%; 1-year OS: p = 0.64, I2 = 0%; 2-year OS: p = 0.21, I2 = 36%). The results of the meta-analysis showed that the 6-month OS (OR = 3.41, 95% CI: 1.62–7.14, p = 0.001), 1-year OS (OR = 1.95, 95% CI: 1.28–2.96, p = 0.002) were higher in the rAd-p53 combination TACE group than in the TACE group. All differences were statistically

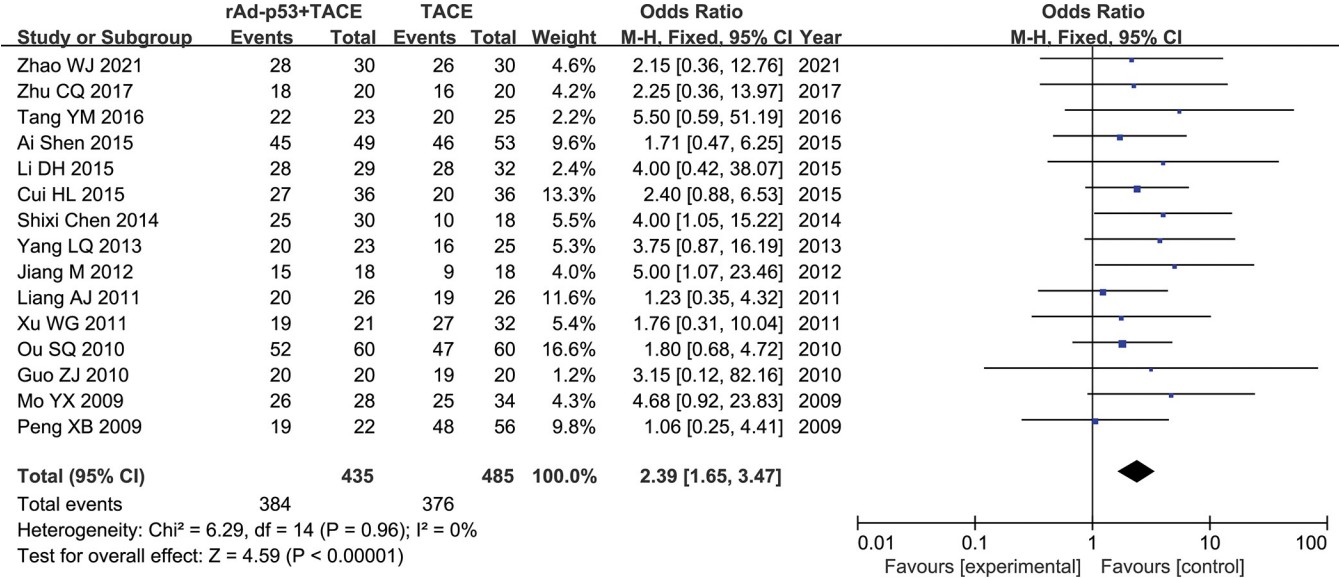

**Fig 6. Forest plot for DCR of rAd-p53 combined with TACE group and TACE alone group.**

**A**

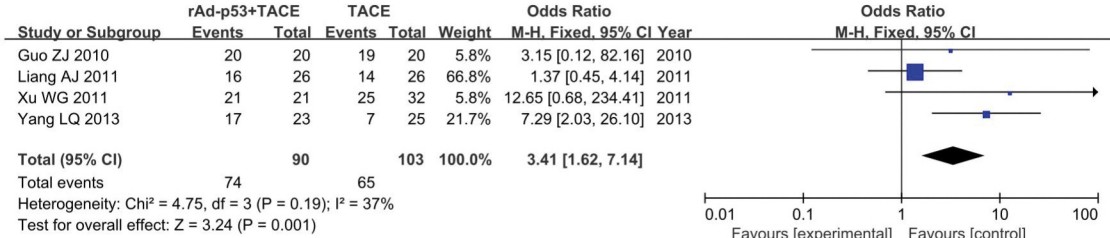

**B**

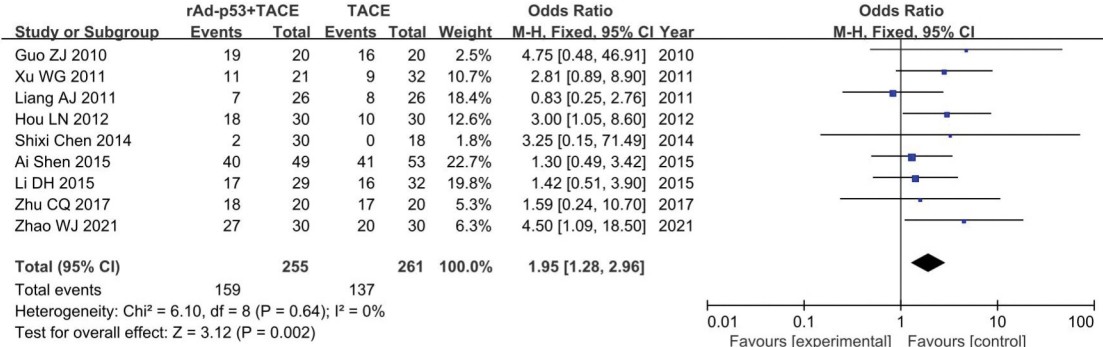

**C**

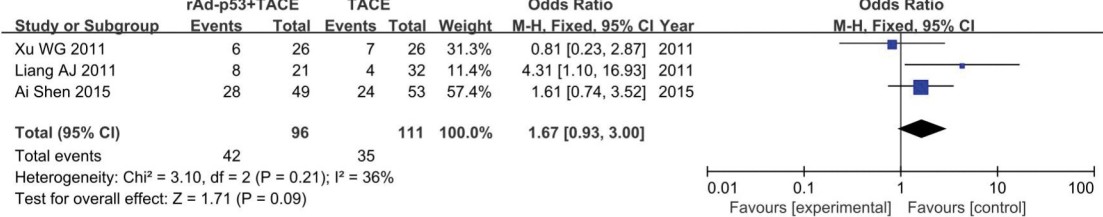

**Fig 7.** Forest plot for six-month (A), one-year (B), and two-year (C) survival rates of rAd-p53 combined with TACE group and TACE alone group.

significant.The difference in 2-year OS (OR = 1.67, 95% CI: 0.93–3.00, p = 0.09) between the two groups was not statistically significant.(Fig 7).

## Quality of life

Six papers [18,20,21,24,25,30] reported changes in KPS scores before and after treatment in the experimental and control groups. There is heterogeneity between studies (p = 0.006, $I^2$ = 69%), therefore, the randomed-effects model was used for the meta-analysis. The results of meta-analysis showed that the improvement of Qol after rAd-p53 combined with TACE treatment was superior to that of the TACE group (MD = 5.84, 95% CI:2.09–9.60, P = 0.002) (Fig 8).

## AFP

Four studies [18,22,24,30] reported changes in AFP after treatment in the experimental and control groups. There is little heterogeneity between studies(p = 0.05, $I^2$ = 62%); therefore, the randomed-effects model was used for the meta-analysis. The results of meta-analysis showed

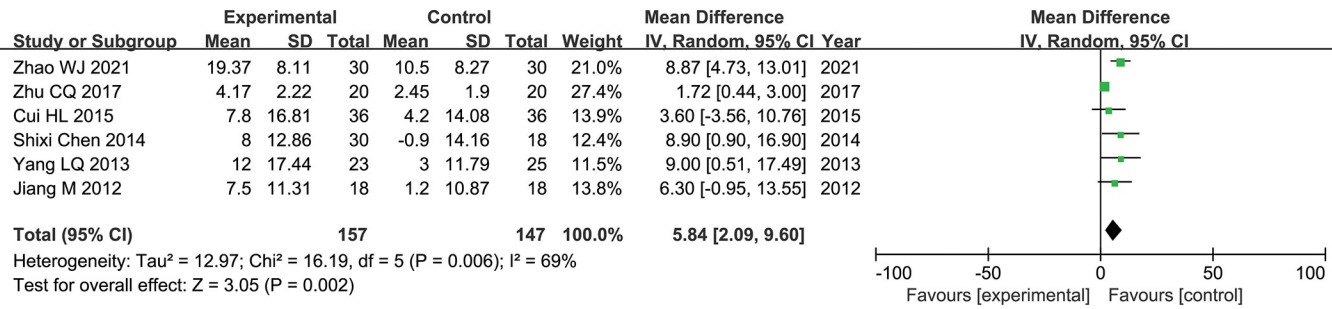

**Fig 8. Forest plot for KPS score of rAd-p53 combined with TACE group and TACE alone group.**

that the levels of AFP after treatment in the rAd-p53 combined with TACE group was lower than that in the TACE group.(MD = -0.64, 95% CI:-1.04–-0.24, P = 0.002) (**Fig 9**).

## Evaluation of patient immunity

Three studies [18,19,24] reported immune status of patients between two groups at 1, 2 and 4 weeks after treatment. The results of the Meta-analysis showed that there was no statistical difference in the percentages of $CD3^+$ cells between the two groups after 1, 2 and 4 weeks of treatment. The percentages of $CD4^+$ cells in the rAd-p53 combined with TACE were higher than those in the TACE group after 2 and 4 weeks of treatment(2week after: MD = 2.05, 95% CI:0.45–3.65, P = 0.01; 4 week after: MD = 2.08, 95% CI:0.43–3.73, P = 0.01). There was a statistically significant difference inpercentages of $CD8^+$ between between the two groups at 1 week(MD = 1.81, 95% CI:0.71–2.91, P = 0.001), The $CD4^+/CD8^+$ ratios of the rAd-p53 combined with TACE were significantly higher than those for TACE group(1week after: MD = 0.11, 95% CI:0.04–0.18, P = 0.002; 2week after: MD = 0.13, 95% CI:0.06–0.20, P = 0.0002; 4 week after: MD = 0.12, 95% CI:0.04–0.20, P = 0.003) (**S1 Fig**, **Table 2**).

## Adverse effects

Seven studies [18,22,23,26,28,29,32] reported fever, the result of meta-analysis showing that the incidence of fever was higher in the rAd-p53 combined with TACE group than in TACE group(OR = 2.62, 95% CI:2.02–3.49, P<0.00001). Six studies [19,23,26,28,31,32] reported myelosuppression, eleven [18,19,22–24,26,28–31,34] studies reported gastrointestinal reactions and two studies [20,29] reported impaired liver function at end of treatment in both groups. According to the results of Meta-analysis, The rates of myelosuppression and gastrointestinal reactions in the rAd-p53 combination TACE group was lower than TACE alone group, The difference was statistically significant. The impaired liver function incidences showed no significant differences between two groups (**Fig 10**).

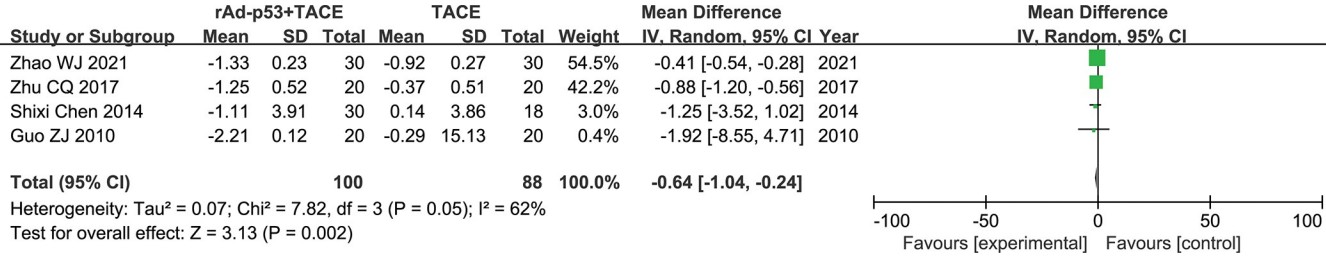

**Fig 9. Forest plot for AFP change of rAd-p53 combined with TACE group and TACE alone group.**

**Table 2. Summary of the results of meta-analysis.**

| Outcomes | | Included study | Number of Exp | Number of Con | Heterogeneity | | Meta-analysis model | Result of meta-analysis | |
|---|---|---|---|---|---|---|---|---|---|
| | | | | | P | I2 | | OR/MD(95%CI) | P |
| CR | | 5 | 127 | 129 | 0.68 | 0% | fixed-effects | 2.19(1.13–4.22) | 0.02 |
| | RCT | 3 | 61 | 63 | 0.43 | 0% | fixed-effects | 2.33(1.05–5.17) | 0.04 |
| | No-RCT | 2 | 66 | 66 | 0.46 | 0% | fixed-effects | 1.93(0.61–6.11) | 0.26 |
| PR | | 15 | 435 | 485 | 0.97 | 0% | fixed-effects | 2.22(1.67–2.94) | <0.00001 |
| | RCT | 7 | 174 | 173 | 0.69 | 0% | fixed-effects | 2.16(1.32–3.51) | 0.002 |
| | No-RCT | 8 | 261 | 312 | 0.97 | 0% | fixed-effects | 2.25(1.59–3.18) | <0.00001 |
| ORR | | 16 | 461 | 524 | 0.92 | 0% | fixed-effects | 2.58(1.95–3.41) | <0.00001 |
| | RCT | 7 | 174 | 173 | 0.53 | 0% | fixed-effects | 3.46(2.01–5.97) | |
| | No-RCT | 9 | 287 | 351 | 0.99 | 0% | fixed-effects | 2.31(1.67–3.19) | |
| DCR | | 15 | 435 | 485 | 0.96 | 0% | fixed-effects | 2.39(1.65–3.47) | <0.00001 |
| | RCT | 7 | 174 | 173 | 0.79 | 0% | fixed-effects | 3.26(1.75–6.07) | |
| | No-RCT | 8 | 261 | 312 | 0.97 | 0% | fixed-effects | 2.00(1.25–3.18) | |
| Month OS | | 4 | 90 | 103 | 0.19 | 37% | fixed-effects | 3.41(1.62–7.14) | 0.001 |
| 1-year OS | | 9 | 255 | 261 | 0.64 | 0% | fixed-effects | 1.95(1.28–2.96) | 0.002 |
| 2-year OS | | 3 | 96 | 111 | 0.21 | 36% | fixed-effects | 1.67(0.93–3.00) | 0.09 |
| KPS | | 6 | 157 | 147 | 0.006 | 69% | random-effects | 5.84(2.09–9.60) | 0.002 |
| AFP | | 5 | 140 | 128 | <0.00001 | 92% | random-effects | -0.98(-1.65–0.31) | 0.004 |
| Toxicity | | | | | | | | | |
| | fever | 7 | 199 | 257 | 0.005 | 68% | fixed-effects | 2.95(1.92–4.53) | <0.00001 |
| | myelosuppression | 6 | 169 | 226 | 0.89 | 0% | fixed-effects | 0.56(0.37–0.86) | 0.008 |
| | gastrointestinal reaction | 11 | 300 | 342 | 0.43 | 1% | fixed-effects | 0.57(0.39–0.81) | 0.002 |
| | impaired liver function | 2 | 65 | 68 | 0.56 | 0% | fixed-effects | 0.70(0.31–1.61) | 0.40 |
| CD3+ | | | | | | | | | |
| | 1 weeks after treatment | 3 | 73 | 75 | 0.25 | 29% | fixed-effects | 2.05(-0.40–4.49) | 0.01 |
| | 2 weeks after treatment | 3 | 73 | 75 | 0.02 | 74% | fixed-effects | 1.21(-1.12–3.54) | 0.31 |
| | 4 weeks after treatment | 3 | 73 | 75 | 0.0002 | 88% | fixed-effects | 1.22(-1.20–3.65) | 0.32 |
| CD4+ | | | | | | | | | |
| | 1 weeks after treatment | 3 | 73 | 75 | 0.37 | 0% | fixed-effects | 0.93(-0.62–2.48) | 0.24 |
| | 2 weeks after treatment | 3 | 73 | 75 | 0.18 | 43% | fixed-effects | 2.05(0.45–3.65) | 0.01 |
| | 4 weeks after treatment | 3 | 73 | 75 | 0.06 | 65% | fixed-effects | 2.08(0.43–3.73) | 0.01 |
| CD8+ | | | | | | | | | |
| | 1 weeks after treatment | 3 | 73 | 75 | 0.45 | 0% | fixed-effects | 1.81(0.71–2.91) | 0.001 |
| | 2 weeks after treatment | 3 | 73 | 75 | 0.77 | 0% | fixed-effects | 0.92(-0.19–2.03) | 0.11 |
| | 4 weeks after treatment | 3 | 73 | 75 | 0.51 | 0% | fixed-effects | 0.78(-0.61–2.17) | 0.27 |
| CD4+/CD8+ | | | | | | | | | |
| | 1 weeks after treatment | 3 | 73 | 75 | 0.70 | 0 | fixed-effects | 0.11(0.04–0.18) | 0.002 |
| | 2 weeks after treatment | 3 | 73 | 75 | 0.61 | 0 | fixed-effects | 0.13(0.06–0.20) | 0.002 |
| | 4 weeks after treatment | 3 | 73 | 75 | 0.71 | 0 | fixed-effects | 0.12(0.04–0.20) | 0.003 |

## Sensitivity analysis and publication bias

We performed a sensitivity analysis by excluding the included studies sequentially from the pooled effect and estimated the effect of each study on the overall outcome by observing the change in outcome after deletion. The results showed that the significance of OR did not change after the exclusion of any of the studies. In addition, we also did a sensitivity analysis plot regarding CR,PR,ORR,DCR and by visual inspection, each study was within the estimated range and had no significant effect on the results (**Figs 12B–15B**), both of which indicate that

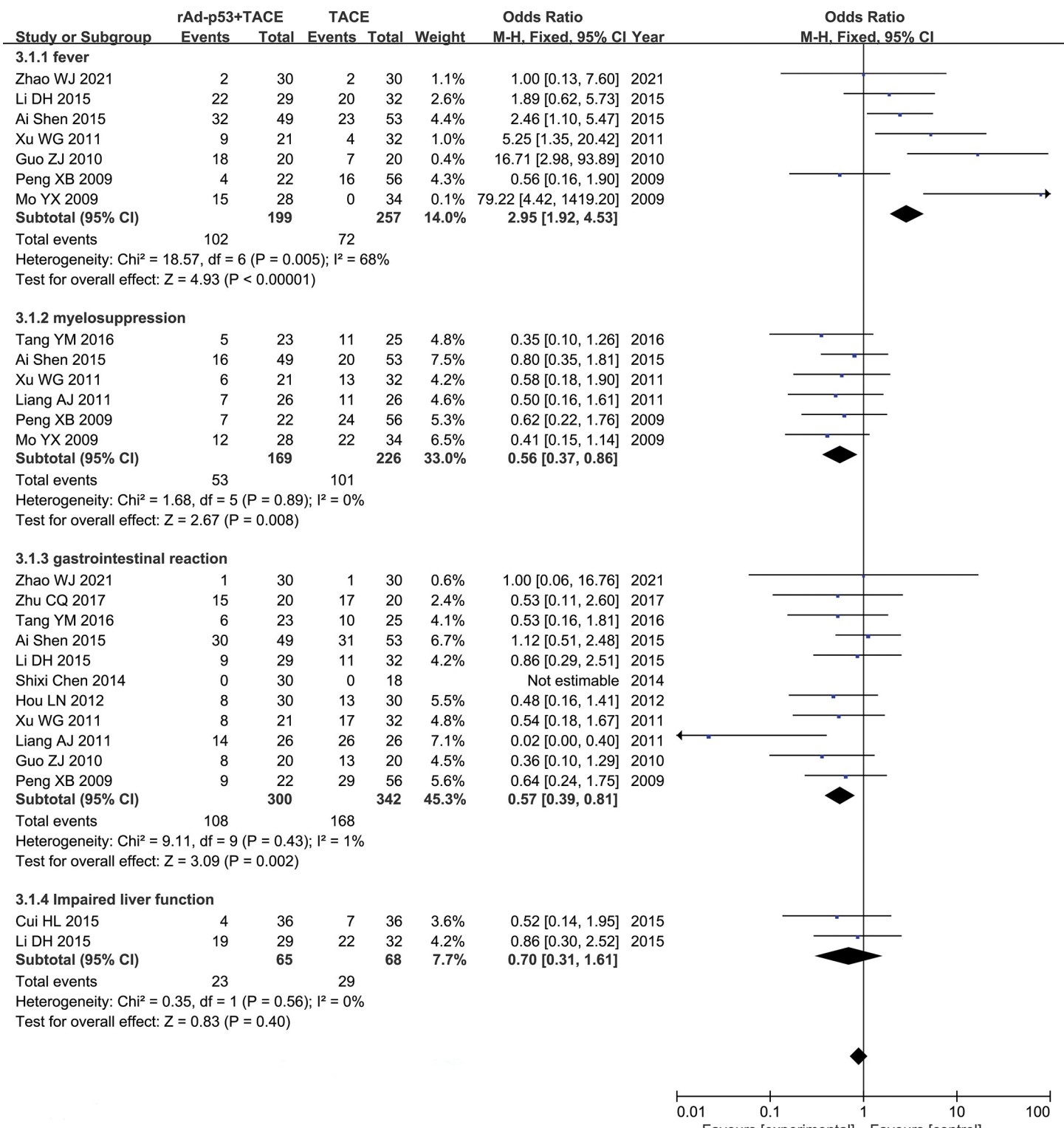

**Fig 10. Forest plot for adverse reactions of rAd-p53 combined with TACE group and TACE alone group.**

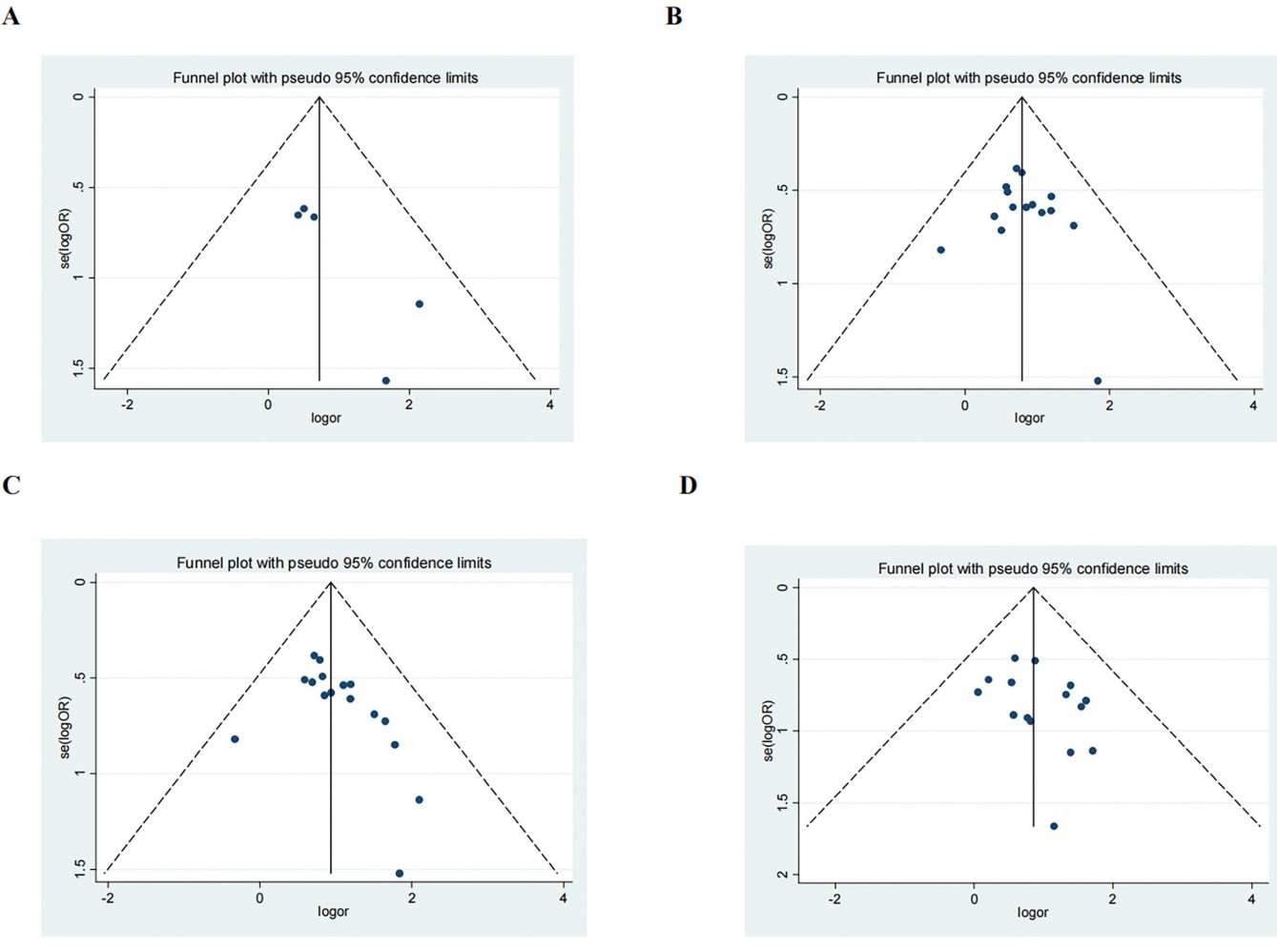

**Fig 11.** Funnel plot of the analysis of CR(A), PR(B),ORR(C), DCR(D).

the results of our meta-analysis are highly reasonable and reliable. We plotted the funnel diagram by stata (**Fig 11**) and performed Begg's and Egger's funnel plots to detect publication bias. The results showed that only the funnel plot of ORR showed publication bias, and we plotted the funnel plot of the cut-and-patch method (**Fig 14A**), which showed that in the future we could eliminate publication bias by adding three more papers. Other funnel plots showed no significant publication bias with p>0.05 (**Figs 12A–13A and 15A**). We also summarize the results of Egger regression tests (**Table 3, S2 Fig**).

## Discussion

Since 1998, 22 gene therapy drugs have been approved for the treatment of human diseases, including nucleic acid, non-viral vectors, viral vectors and cell-mediated gene therapy products [35]. Since most cancers have abnormalities in the p53 gene signaling pathway, inactivation of the p53 gene allows tumor cells to escape apoptosis and is strongly associated with malignancy development, prognosis, and resistance to radiotherapy, all of these factors contribute to make the tumor suppressor gene P53 a promising target for gene therapy [36].The concept of how to put wild type (WT) p53 back into cancer cells leads to the regression of tumor cells has facilitated the development of recombinant adenoviruses carrying wild p53

A

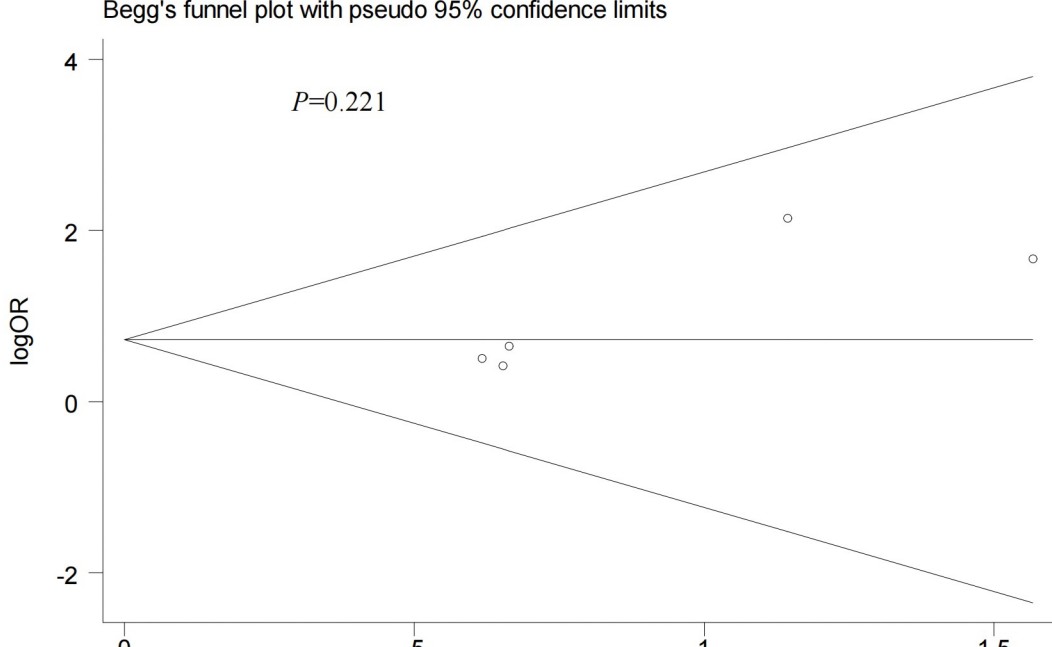

B

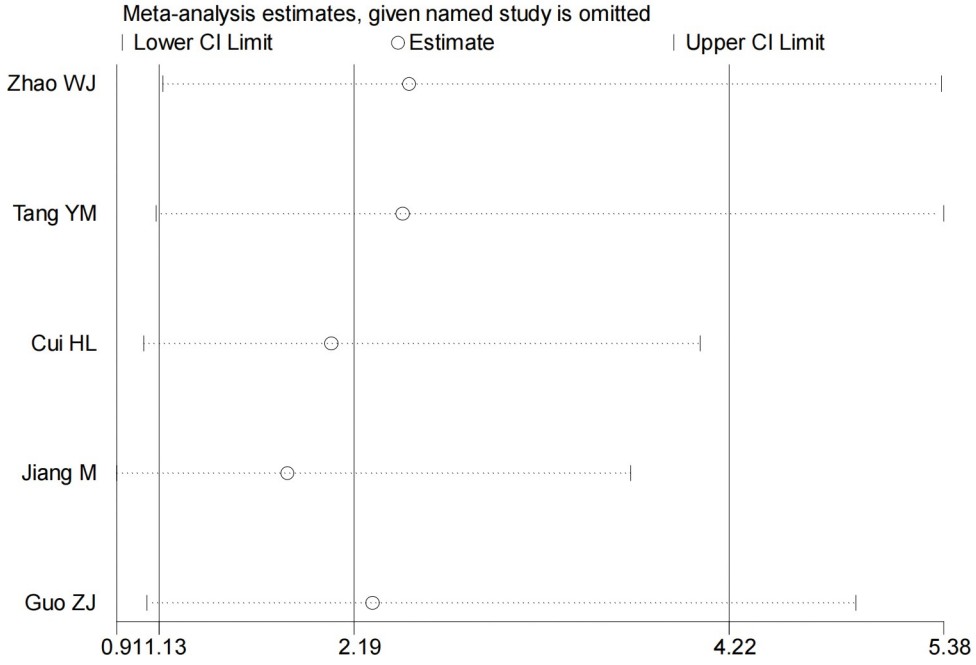

**Fig 12.** Begg's funnel plot (A) and sensitivity analysis (B) of CR. Begg's test (P = 0.221).

**A**

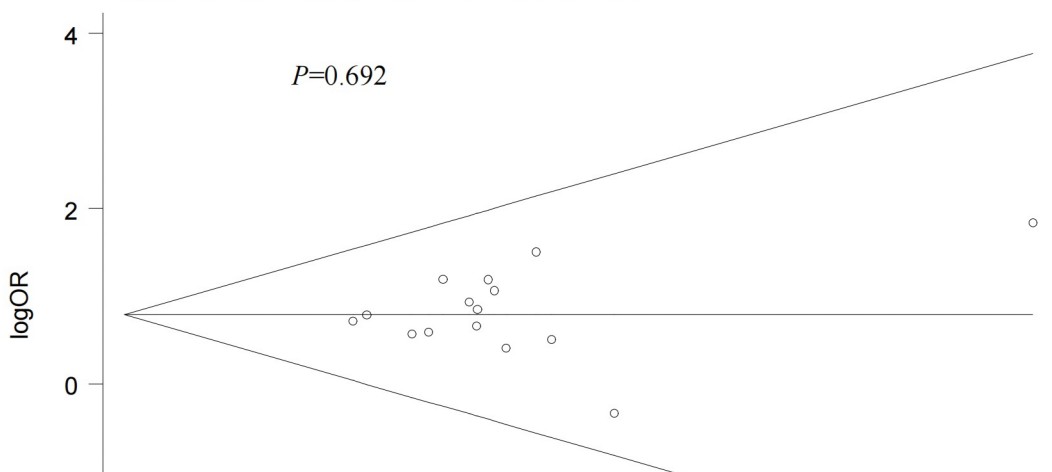

**B**

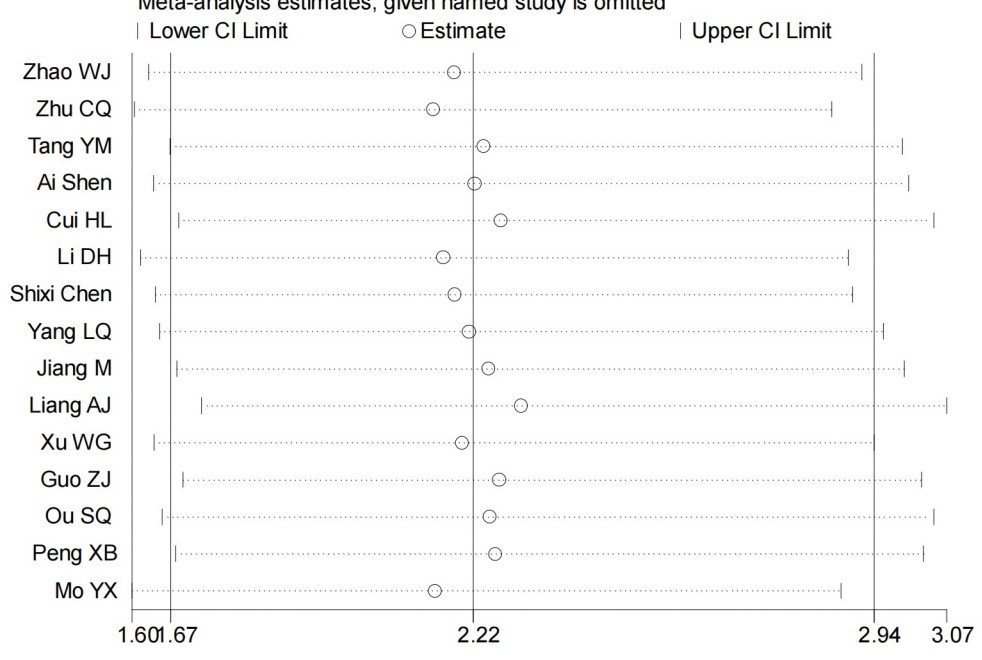

**Fig 13.** Begg's funnel plot (A) and sensitivity analysis (B) of PR. Begg's test (P = 0.692).

**A**

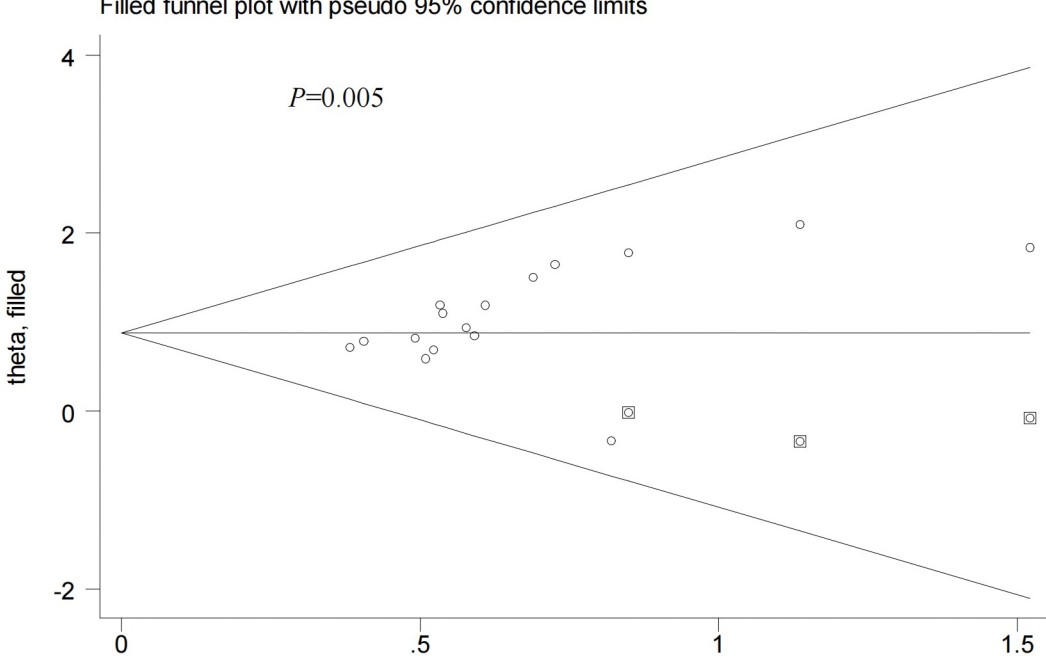

**B**

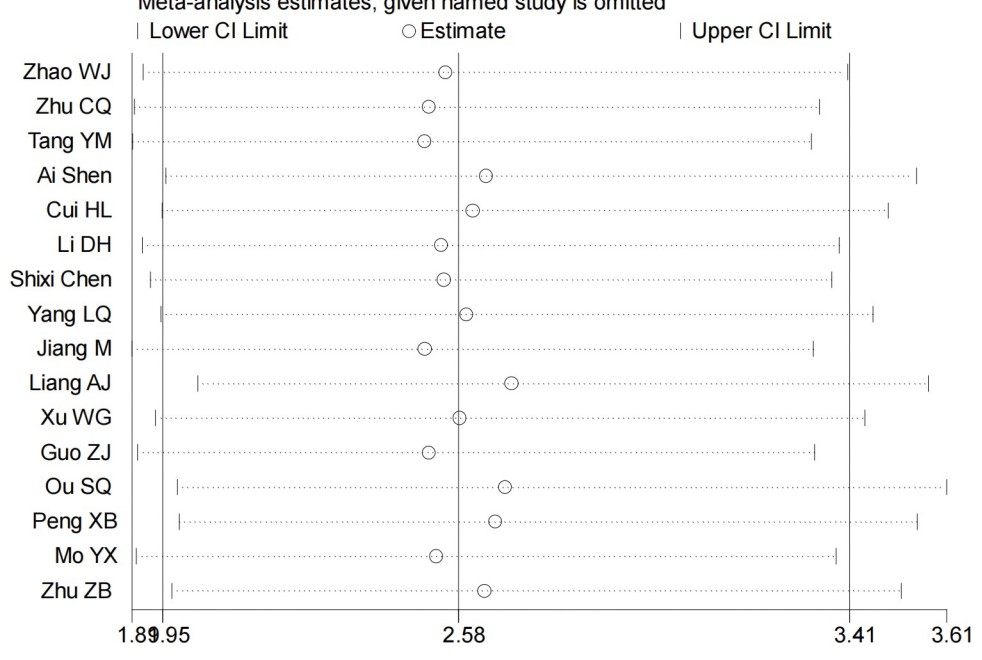

**Fig 14.** Begg's funnel plot (A) and sensitivity analysis (B) of ORR.Begg's test (P = 0.005).

**A**

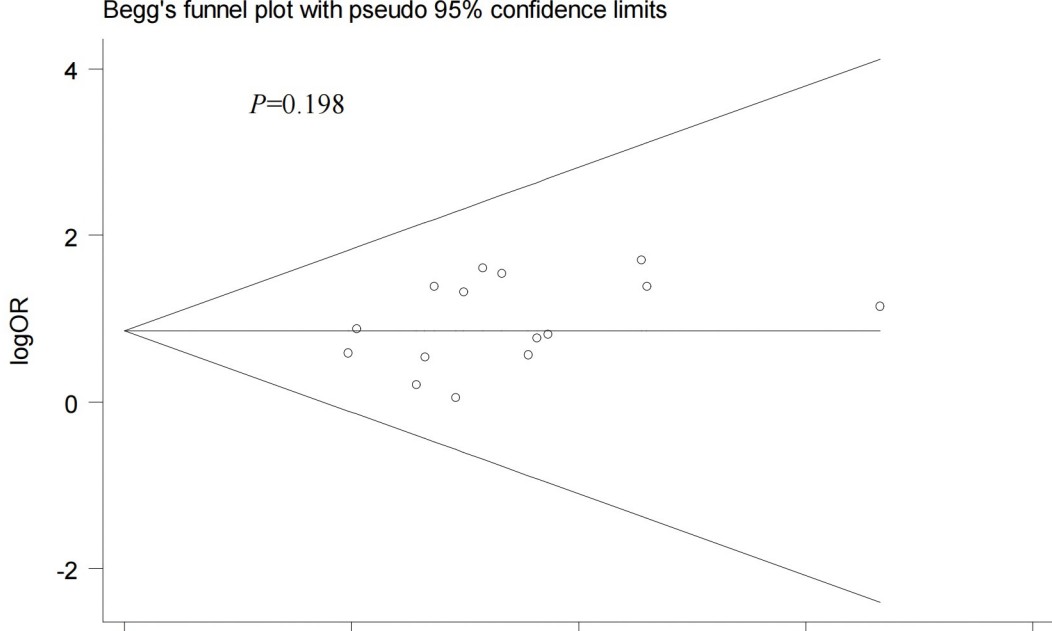

**B**

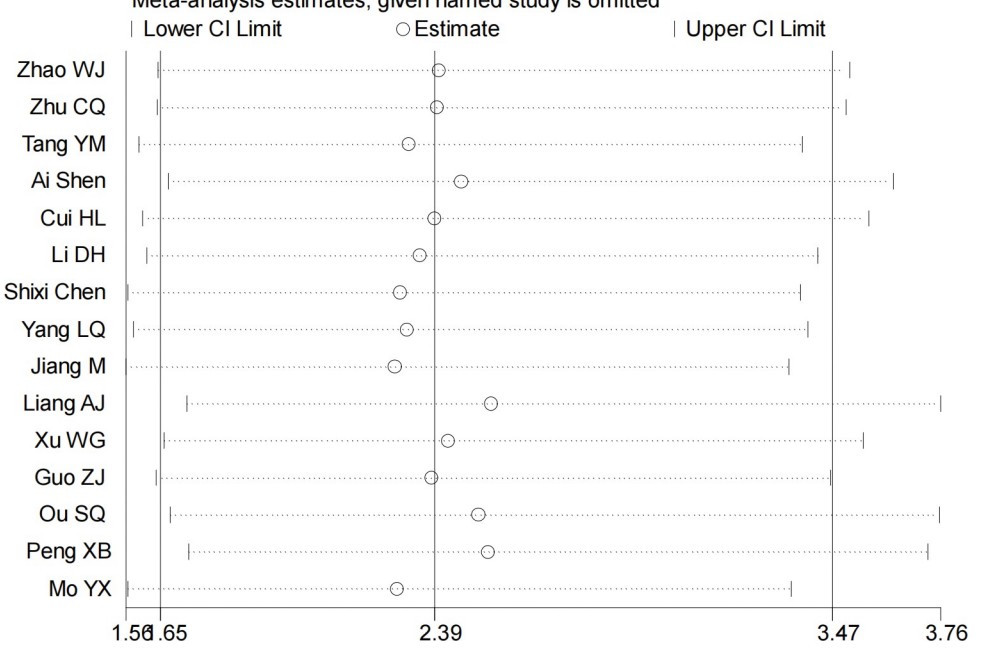

**Fig 15.** Begg's funnel plot (A) and sensitivity analysis (B) of DCR.Begg's test (P = 0.198).

**Table 3. Result of Egger's regression tests.**

| Evaluation Items | t | 95%CI | | P |
|:---:|:---:|:---:|:---:|:---:|
| CR | 3.09 | -2.181 | 0.837 | 0.221 |
| PR | 0.46 | -1.109 | 1.709 | 0.692 |
| ORR | 2.13 | 2.13 | -0.009 | 0.005 |
| DCR | 1.46 | -0.418 | 2.165 | 0.198 |

genes. The basic mechanism of rhAd-p53 reagent is as follows [37]: Inhibiting tumor growth by cell cycle arrest and inducing programmed cell death; (ii) Inhibiting tumor growth by cell cycle arrest and inducing programmed cell death; (ii) Enhance chemotherapy-induced cell cycle arrest and apoptosis; (iii) Stimulate the body to produce anti-tumor immunity, such as a large number of immune cells gathered in the local injection site of the tumor. (iv) Inhibition of tumor vascular endothelial growth factor (VEGF) to suppress angiogenesis and tumor growth through the "bystander effect"; therefore, the local tumor tissue at the injection site will block the blood supply and induce tumor necrosis [38].

In 2003, the National Medical Products Administration (NMPA) of China approved the first anti-tumor gene therapy drug (rAd-p53) in combination with radiotherapy for the treatment of Head and Neck Squamous Cell Carcinoma (HNSCC) [39].The main principle of rAd-p53 is to introduce the P53 gene into cancer cells by constructing an adenoviral vector to express the p53 protein and perform biological functions. Adenovirus was chosen as the gene vector because it can infect a variety of cell types, including dividing and non-dividing cells; it does not integrate vector DNA into host cells; it has a high level of transgene expression; it supports high-titer, large-scale production; and the safety of adenovirus vaccines has been established [40,41]. In the 12 years since its launch in 2004, Gendicine has published results from more than 30 clinical trials of rAd-p53 and rAd-p53 in combination with other therapeutic regimens for the treatment of HNSCC and other cancers, and to date, Gendicine has been used clinically in China to treat more than 30,000 patients [42]. Hepatocellular carcinoma is the second most common type of cancer treated with the rAd-p53 combination. There have been many published clinical trials on rAd-p53 for the treatment of hepatocellular carcinoma, and good clinical efficacy has been achieved. The purpose of this study was to systematically evaluate the clinical efficacy survival and complications of Rad-p53 combined with TACE in the treatment of hepatocellular carcinoma, and to provide evidence-based medical evidence to improve the clinical outcomes of patients with hepatocellular carcinoma.

Data from 17 clinical trials including a total of 1045 patients were used in our meta analysis. The results of our meta-analysis showed that the rAd-p53 combined with TACE group had significantly improved CR, PR and ORR,DCR compared with TACE alone. We did a subgroup analysis based on whether it was an RCT trial, and the results of the subgroup analysis were consistent with the overall(**Sig S3, Table 2**). In terms of long-term survival, the rAd-p53 combined with TACE group had a better 6-month OS and 1-year OS than the control group. A total of three studies reported 2-year OS, and meta-analysis of 2-year OS showed no statistical significance, but each of the included studies had better 2-year OS in the combination therapy group than in the control group.One study [23] in our included studies also reported progression-free survival(PFS) in both groups, the result showed that the rAd-p53-based TACE treatment strategy improved the OS (HR: 0.58, 95%CI: 0.35–0.96, P = 0.035), PFS(HR: 0.60, 95% CI: 0.37–0.97, P = 0.037). And the quality of life of rAd-p53 combined with TACE for hepatocellular carcinoma patients was also higher than that of the control group. Recent studies have utilized AFP for guiding therapeutic decisions and prognostic analysis and monitoring HCC

treatment [43].The results of our meta-analysis showed that AFP levels in patients after rAd-p53 combined with TACE treatment were lower than those in the control group.

In adverse reactions, the incidence of fever was higher in the rAd-p53 combined with TACE group than in the control group.As is common knowledge, gene therapy experiments commonly involve viral vectors—particularly adenovirus—causing strong immune reactions against the vector [44],usually presents as a self-limiting fever that appears within 24 hours of injection. grade II fever resolves on its own, while grade III fever returns to normal after administration of antipyretic medication.To our surprise the remaining adverse effects myelo-suppression and gastrointestinal reaction rates were lower than in the TACE alone group and there was no difference in liver function impairment between the two groups.Thus, rAd-p53 appears to be a safe genetic agent for patients with hepatocellular carcinoma, which could alle-viate some of the adverse events associated with conventional treatment.

A point of interest is that the three studies we included examined changes in T lymphocyte subsets (CD3+, CD4+, CD8+ cell subsets and CD4+/CD8+ ratio) which play an important role in antitumor immunity after treatment in both groups of patients. Our analysis showed that the percentages of CD4+ and the CD4+/CD8+ ratio were increased in the rAd-p53 com-bined with TACE group, suggesting that the immune function of liver cancer patients was improved after rAd-p53 treatment, so we believe that rAd-p53 also has great potential in the immunotherapy of liver cancer. Clinical studies of rAd-p53 in combination with immune checkpoint inhibitors have been reported. A study on the induction of antitumor activity of rAd-p53 combined with immune checkpoint inhibitor (ICI) anti-PD-1 in a mouse homolo-gous urogenital cancer model was reported in 2020 [25] and found that the combination of rAd-p53 and anti-PD-1 induced tumor-infiltrating T cells and that rAd-p53 increased PD-L1 expression in tumor cells in vivo. A phase 2, multicenter, open study initiated by the Robert H. Lurie Comprehensive Cancer Center in Chicago to evaluate a clinical trial of rAd-p53 in com-bination with an immune checkpoint inhibitor anti-PD-1 or anti-PD-L1 in patients with recurrent or metastatic head and neck squamous cell carcinoma and other tumors (www.clinicaltrials.gov, NCT03544723). The trial is ongoing and will end on November 31, 2022, and its findings are highly anticipated. Further studies are needed in the future to explore the potential of rAd-p53 in cancer immunotherapy.

## Limitations

Our meta-analysis has several limitations. First, the doses of rAd-p53 used in the included studies were not standardized, and there were 2 studies for which we could not obtain infor-mation on TACE regimens. Second, a portion of the studies did not have long-term follow-up, and we speculate that the study endpoints may not have been met. More studies are needed to assess the relationship between rAd-p53 combined with TACE and long-term survival in hepatocellular carcinoma, and we will continue to monitor the reports of studies in this area in the future. Third, the sample size of the included studies was small, and large multicenter ran-domized clinical trials on rAd-p53 and TACE for liver cancer are still needed in the future to guide us to better treat liver cancer.

## Conclusion

In conclusion, findings of this meta-analysis indicate showed that rAd-p53 combined with TACE for liver cancer improved clinical outcomes and survival, and effectively improved patient quality of life and immune function. Therefore, our study will provide valuable evi-dence for further evaluation of rAd-p53 in the treatment of liver cancer. On the other hand, considering that only a few clinical trials have evaluated the long-term efficacy and

immunomodulatory effects of rAd-p53, more studies with high-quality evidence are needed to validate the effectiveness of rAd-p53 in the treatment of liver cancer.

## Supporting information

**S1 Appendix. Pubmed literature search strategies.**
(DOCX)

**S1 Fig. Forest plot of immune status of patients in rAd-p53 combined with TACE group and TACE group.** CD3[+] (A), CD4[+](B), CD8[+](C), CD4[+]/CD8[+](D).
(DOCX)

**S2 Fig.**  Begg's and Egger's tests for publication bias of CR(A), PR(B), ORR(C), DCR(D).
(DOCX)

**S3 Fig.**  Forest plot for subgroup analysis of CR(A), PR(B), ORR(C) and DCR(D) based on RCT/Non-RCT.
(DOCX)

**S1 Table. PRISMA checklist.**
(DOC)

## Author Contributions

**Data curation:** Gaolei Ma, Chen Liu, Juan Yang.

**Software:** Yaru Guo, Yingnan Zhang, Wenwen Guo, Jingya Zhang.

**Supervision:** Xiaojin Wu.

**Writing – original draft:** Yaru Guo.

**Writing – review & editing:** Yuanyuan Chen, Mengjun Xu.

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
