## [Decision Letter · Decision Letter 0]

17 May 2023

PONE-D-23-04833Recombinant human adenovirus p53 combined with transcatheter arterial chemoembolization for liver cancer: a meta-analysisPLOS ONE

Dear Dr. xiaojin,

Thank you for submitting your manuscript to PLOS ONE. After careful consideration, we feel that it has merit but does not fully meet PLOS ONE’s publication criteria as it currently stands. Therefore, we invite you to submit a revised version of the manuscript that addresses the points raised during the review process.

Dear Authors, please respond to all reviewers' comments, also provided by the reviewer that had rejected the paper. Please kindly focus also on comments regarding study methodology, as it reguires in depth revision.

We look forward to receiving your revised manuscript.

Kind regards,

Mirosława Püsküllüoğlu, MD, PhD

Academic Editor

PLOS ONE

Journal Requirements:

Reviewers' comments:

Reviewer's Responses to Questions

**Comments to the Author**

1. Is the manuscript technically sound, and do the data support the conclusions?

Reviewer #1: Yes

Reviewer #2: Yes

Reviewer #3: No

2. Has the statistical analysis been performed appropriately and rigorously? 

Reviewer #1: Yes

Reviewer #2: I Don't Know

Reviewer #3: Yes

3. Have the authors made all data underlying the findings in their manuscript fully available?

Reviewer #1: Yes

Reviewer #2: Yes

Reviewer #3: Yes

4. Is the manuscript presented in an intelligible fashion and written in standard English?

Reviewer #1: Yes

Reviewer #2: Yes

Reviewer #3: Yes

5. Review Comments to the Author

Reviewer #1: The study is of interest and a reasonable number of studies were included. I have the following comments:

1) Since the number of included studies is high, i would recommend to restrict the main analysis only to RCTs.

2) Were response criteria the same in the included studies? This is quite hard to believe. The same for timing of response assessment.

3) Was the technique used the same? These aspects should be better explained in the material and methods part.

4) Time to event outcomes should be expressed in terms of HR , not OR.

5) Quality of life represents a variable too heterogeneous (different scale for assessment, lack of blinding). I would recommend to remove it.

6) AFP analysis is too heterogeneous. The authors should try to explore the eventual sources of heterogeneity.

7) AE analysis was completely lacking. This represents an important outcome that can have also influence on the prognosis (cite and comment the recent paper PMID: 34683182)

8) English grammar should be improved.

9) The authors should comment the current state of art of HCC management (cite the recent paper PMID: 33339274)

Reviewer #2: I would like to thank you very much for the opportunity to review the manuscript titled "Recombinant human adenovirus p53 combined with transcatheter arterial chemoembolization for liver cancer: a meta-analysis"

1. It is sound more applicable to write HCC etiologies and stratify patients as well as treatment approach accordingly.

2. NK cells have an important role in HCC; therefore, it is advised to include their potential anti-tumor role and activation tests.

3. The authors should cite the mechanism of action of the medication on cancer related to signaling pathways of survival.

4. The patients’ number included in the article is low 1045 as compared to the patients’ number in the studies.

Reviewer #3: The authors carry of a meta-analysis comparing stadard TACE therapy vs TACE therapy combined with rAv-p53. They included 17 studies with 1045 patiients were evaluated (554 vs 491 in each group). They find that combined therapy improved co mplete remission rate, partial remission, TRR and improved survival benefit at 6 months/1 year, associated with some effects of patient immunity.

There are a number of issues in this manuscript.

1. They included 8 RCTs and 9 non-RCTs which risks selection bias and heterogeneity.

2. Whilst they used the search terms on multiple data bases they provide no information from which data bases the included studies were taken. They may all have come from one data base.

3. There are no data on tumor staging/characterisation which given the nature of included studies risks further selection bias.

4. As the authors note the rAv-p53 regimens are non-standardized as infact are the agents used in the TACE regimens.

Overall the manuscript would benefit from some minor edits for clarity.

6. PLOS authors have the option to publish the peer review history of their article (what does this mean?). If published, this will include your full peer review and any attached files.

Reviewer #1: No

Reviewer #2: No

Reviewer #3: No

---

## [Author Response · Author response to Decision Letter 0]

8 Jul 2023

Answer to the question of reviewer one

Question 1：Since the number of included studies is high, i would recommend to restrict the main analysis only to RCTs.

My answer：Thank you for your suggestion, We once again carefully read the articles we included. Our criteria for inclusion in the literature were that clinical trials on rAd-p53 combined with TACE for hepatocellular carcinoma are prospective studies, and retrospective studies have been excluded. However, the specificity of the rAd-p53 treatment modality in the experimental group and the informed consent of the patients included in the trials resulted in some clinical trials not taking a randomized control approach to group the number of participants, so we included all trials that met the final results.The results of the meta analysis indicated that the the rAd-p53combined with TACE markedly improved the patients’ complete remission(OR=2.19, 95% CI:1.13–4.22, P=0.02), partial remission (OR=2.22, 95% CI:1.67–2.94, P<0.00001), objective tumor response rate (OR=2.58, 95% CI:1.95–3.41, P<0.00001) and disease control rate(OR=2.39, 95% CI:1.65–3.47, P<0.00001) compared with TACE alone. Your proposal to restrict the primary analysis to RCT studies is, in our opinion, very correct, and we considered this issue at the time when we analyzed the data. The essence of meta-analysis is the discipline of aggregating all studies that fit a particular subject study and finally performing a statistical analysis to draw a definitive conclusion. In order to make our results more reliable, we included all the studies that fit. However, in order to exclude whether the inclusion of studies that were RCTs would have an effect on our experimental results, We did a subgroup analysis based on whether it was an RCT trial, and the results of the subgroup analysis were consistent with the overall，and There was nosignificant heterogeneity between studies (p = 0.96, I2 = 0%).The specific results can be seen in our manuscript in S3 Fig, Table 2。

Question 2：Were response criteria the same in the included studies? This is quite hard to believe. The same for timing of response assessment.

My answer：

Thank you very much for your valuable comments. We can tell you for sure that the evaluation criteria for the study results are the same.The primary efficacy outcomes were complete remission(CR), partial remission (PR), objective tumor response rate (ORR) and disease control rate(DCR) According to the internationally accepted criteria for evaluating the efficacy of solid tumors (RECIST 1.1).CR:Complete disappearance of the lesion, no new lesion creation, and lasting for more than 4 weeks.PR:The sum of the maximum diameter of the lesions was reduced by more than 30% and lasted for more than 4 weeks.ORR ORR was calculated as follows: (CR + PR)/total number of cases×100%. DCR:The ORR was calculated as follows: (CR + PR+SD)/total number of cases×100%.Adverse reactions in all clinical trials were evaluated and graded according to the acute adverse reaction evaluation criteria CTCAE5.0 for hematological toxicity, gastrointestinal reactions and hepatic and renal impairment as side effects that occurred during patient treatment. The duration of reaction assessment was essentially 4 weeks. This is reflected in the included literature, and we are very sorry that it is not reflected in our article.We have been in the revision of the red mark in the engagement, please check, thank you very much for your valuable advice to us, is our oversight.

Question 3:Was the technique used the same? These aspects should be better explained in the material and methods part.

My answer：The technology used is the same, and the technology used is TACE.Under local anesthesia, a 5 F hepatic tube was punctured through the femoral artery and placed in the common hepatic artery and superior mesenteric artery for imaging to clarify the number, location, size, blood supply vessels and the presence of arteriovenous fistula of the tumor. Then, microcatheters were used to superselect to the tumor target vessels, and epirubicin, cisplatin and 5-Fu were injected mainly via catheter to perform embolization chemotherapy; in the experimental group, after embolization and embolization of chemotherapeutic drugs, recombinant human p53 adenovirus injection (rAdp53, Shenzhen Sebenogene Technology Co., Ltd, aqueous injection form, size 1×1012VP/stem. The drug was stored at -20℃ and melted at room temperature before use) was infused.We apologize for not reflecting this in the original article, we have revised the approach to reflect it in the draft, thank you very much for your valuable comments.

Question 4:Time to event outcomes should be expressed in terms of HR , not OR.

My answer：

Thank you for your suggestion on our statistical results, after our further study we felt that OR was more appropriate, we mainly studied the effect of Rad-P53 (exposure factor) on the experimental results. OR is used to reflect the difference between cases and controls in terms of exposure, thus establishing a link between disease and exposure factors so OR was finally chosen, mainly to study the difference in the number of events (efficiency, side effects, six-month, one-year, two-year survival) in the Rad-p53 group compared to the non-Rad-p53 group. HR is the risk function ratio, which is a multiple used in survival analysis data to estimate the change in risk of death/remission/recurrence due to the presence of a factor. In summary, we think OR is more appropriate and thank you very much for your suggestion to deepen our learning on the representation of statistical outcomes.

Question 5:Quality of life represents a variable too heterogeneous (different scale for assessment, lack of blinding). I would recommend to remove it.

My answer：

Thank you very much for your suggestion, it was an oversight on our part that we did not show the specific details of the KPS score in the original article. In our included literature, a total of 6 papers compared the change in life treatment before and after treatment in the experimental and control groups. All of the included literature evaluated patients' quality of life according to the Karnofsky scale: patients were divided into ten levels according to their condition and the degree of normal activity and self-care, with 10 points/level and a total score of 100, and patients' health status and quality of life were positively correlated with the score.Six papers(17, 19, 20, 23, 24, 29) reported changes in KPS scores before and aftertreatment in the experimental and control groups. There is heterogeneity betweenstudies (p = 0.006, I2 = 69%), therefore, the randomed-effects model was used for themeta-analysis. The results of meta-analysis showed that the improvement of Qol afterrAd-p53 combined with TACE treatment was superior to that of the TACE group(MD=5.84, 95% CI:2.09–9.60, P=0.002) (Fig 8).

Question 6 :AFP analysis is too heterogeneous. The authors should try to explore the eventual sources of heterogeneity.

My answer：Dear Editor, Regarding the treatment of AFP in the paper, we would like to explain that we are mainly confirming that the reduction of AFP in the experimental group after treatment is greater than the reduction of AFP in the control group after treatment. And in the literature that we included, four papers reported the values of AFP before and after treatment, and AFP is given as the mean value of APF before and after treatment in the experimental and control groups before and after treatment. For the statistical approach of meta-analysis of continuous type data, we have combined the mean mean, standard deviation sd (before and after values are subtracted and then mean and standard deviation are found) before and after treatment in both treatment groups. The specific formula: S*S=S1*S1+S2*S2-2*R*S1*S2 .

Question 7: AE analysis was completely lacking. This represents an important outcome that can have also influence on the prognosis (cite and comment the recent paper PMID: 34683182)

Dear reviewers, we have done an analysis of the relevant complications in the ADVERSE section of the manuscript. Seven studies(17, 21, 22, 25, 27, 28, 31) reported fever, the result of meta-analysis showing that the incidence of fever was higher in the rAd-p53 combined with TACE group than in TACE group(OR=2.62, 95% CI:2.02–3.49, P<0.00001). Six studies(18, 22, 25, 27, 30, 31) reported myelosuppression, eleven(17, 18, 21-23, 25, 27-30, 33) studies reported gastrointestinal reactions and two studies(19, 28) reported impaired liver function at end of treatment in both groups. According to the results of Meta-analysis, The rates of myelosuppression and gastrointestinal reactions in the rAd-p53 combination TACE group was lower than TACE alone group, The difference was statistically significant. The impaired liver function incidences showed no significant differences between two groups. (Fig 10 ).all other adverse effects were lower in the TACE combined with rAd-p53 group than in the TACE group.

Question 9：The authors should comment the current state of art of HCC management (cite the recent paper PMID: 33339274)

My answer：Thank you very much for your suggestion, we have read the study carefully and got a lot out of it, we have cited this article in the preface section, thank you very much for your suggestion.

Finally, once again, thank you for your hard work in reviewing our article and giving us valuable comments. We have carefully revised our article according to your comments, please review it again and if you have any questions, please feel free to contact us and look forward to receiving your reply soon.

Answer to the question of reviewer two

Question 1： It is sound more applicable to write HCC etiologies and stratify patients as well as treatment approach accordingly.

My answer：Thank you very much for your comments on our article, and your suggestions are of great value to us. The aim of this study was to systematically evaluate the clinical efficacy, survival and complications of rAd-p53 combined with TACE for hepatocellular carcinoma through meta-analysis. For the etiology of HCC, we are very sorry that we did not reflect it in the original article, and considering the length of the preface, we only this wrote about the incidence and treatment of hepatocellular carcinoma in recent years. Thank you very much for your suggestion.

Question 2：  NK cells have an important role in HCC; therefore, it is advised to include their potential anti-tumor role and activation tests.

My answer：We think your suggestion is very good and has great research value. However, our meta-analysis is to incorporate the results of previous clinical trials that have been done well, to conduct statistical analysis, to draw a conclusion, and then to provide evidence-based medical evidence for the clinical treatment of liver cancer, so our meta-analysis is actually a secondary study in essence. We reviewed the relevant literature and found that NK cells do play a huge role in hepatocellular carcinoma. In our meta-analysis, we found that immune cells were increased in the Rad-p53 combination treatment group after treatment than in the control group after treatment, and it is unfortunate that we did not further isolate and analyze these enlarged immune cells in this literature for inclusion. In the future, we hope to have more experimental areas to study the effect of RAD-P53 on immune cells in patients with hepatocellular carcinoma, after conducting further analysis of the types of immune cells, and in the future we hope to engage in research in this area, thank you very much for your suggestions, your suggestions provide a good idea for our next work.

Question 3： The authors should cite the mechanism of action of the medication on cancer related to signaling pathways of survival.

My answer：Thank you very much for your suggestion to make us realize our oversight about the mechanism of Rad-p53's anticancer effect, which we should reflect in the article. The basic mechanism of rhAd-p53 reagent is as follows : inhibiting tumor growth by cell cycle arrest and inducing programmed cell death; ( ii ) inhibiting tumor growth by cell cycle arrest and inducing programmed cell death; ( ii ) Enhance chemotherapy-induced cell cycle arrest and apoptosis; ( iii ) Stimulate the body to produce anti-tumor immunity, such as a large number of immune cells gathered in the local injection site of the tumor. ( iv ) Inhibition of tumor vascular endothelial growth factor (VEGF) to suppress angiogenesis and tumor growth through the “bystander effect”;therefore, the local tumor tissue at the injection site will block the blood supply and induce tumor necrosis.We have added this section to the discussion section of the original article and thank you again for your comments.

Question 4: The patients’ number included in the article is low 1045 as compared to the patients’ number in the studies.

My answer：Thank you very much for your suggestion, and we recognize this issue; the Meta-analysis is essentially a secondary study, and the small sample size of the studies we included resulted in a small number of patients in our article, 1045, which we also addressed in the original article in terms of limitations. Future multicenter, large-sample randomized controlled clinical trials are still needed to further investigate this aspect, and we will closely follow the progress in this area.

Finally, once again, thank you for your hard work in reviewing our article and giving us valuable comments. We have carefully revised our article according to your comments, please review it again and if you have any questions, please feel free to contact us and look forward to receiving your reply soon.

Answer to the question of reviewer three

Question 1: They included 8 RCTs and 9 non-RCTs which risks selection bias and heterogeneity.

My answer：Thank you for your suggestion, The specificity of the rAd-p53 treatment modality in the experimental group and the informed consent of the patients included in the trials resulted in some clinical trials not taking a randomized control approach to group the number of participants, so we included all trials that met the final results. The results of the meta analysis indicated that the the rAd-p53combined with TACE markedly improved the patients’ complete remission(OR=2.19, 95% CI:1.13–4.22, P=0.02), partial remission (OR=2.22, 95% CI:1.67–2.94, P<0.00001), objective tumor response rate (OR=2.58, 95% CI:1.95–3.41, P<0.00001) and disease control rate(OR=2.39, 95% CI:1.65–3.47, P<0.00001) compared with TACE alone. There was nosignificant heterogeneity between studies. The essence of meta-analysis is the discipline of aggregating all studies that fit a particular subject study and finally performing a statistical analysis to draw a definitive conclusion. In order to make our results more reliable, we included all the studies that fit. However, in order to exclude whether the inclusion of studies that were RCTs would have an effect on our experimental results, We did a subgroup analysis based on whether it was an RCT trial, and the results of the subgroup analysis were consistent with the overall，and There was nosignificant heterogeneity between studies .The specific results can be seen in our manuscript in S3 Fig, Table 2.

Question 2:Whilst they used the search terms on multiple data bases they provide no information from which data bases the included studies were taken. They may all have come from one data base.

My answer：We comprehensively searched for clinical studies of rAd-p53 combined with TACE for liver carcinoma in the PubMed, the Cochrane of Library, Web of Science, Embase, Wanfang Data, Chinese National Knowledge Infrastructure (CNKI), Chinese Biological Medicine (CBM) Database, and VIP Database until August 2022.In Fig 1 of the original article, we list the final retrieved studies in each data separately. the PubMed (24), the Cochrane of Library (6), Web of Science (44), Embase (97), Wanfang Data (288), Chinese National Knowledge Infrastructure (119), CBM (70).

Question 3: There are no data on tumor staging/characterisation which given the nature of included studies risks further selection bias.

My answer：Thank you very much for your suggestion, we think your suggestion is very good, it was an oversight on our part not to show the staging of the included patients in the original article, the patients included in the study basically belonged to the late stage of the bureau or advanced stage, the patients could not have surgery for various reasons and had to choose to do TACE.

Question 4：As the authors note the rAv-p53 regimens are non-standardized as infact are the agents used in the TACE regimens.

My answer：Thank you very much for your suggestion, this suggestion of yours is very valuable to us, and our limitations in the original article also mentioned this issue, the dose of rAd-p53 used in the included studies was not standardized, the dose of rAd-p53 was determined according to the physical condition of the patient, the size of the tumor, and the number of tumors. Also the TACE regimen, in the studies we included, was basically epirubicin, cisplatin, and 5-Fu based. Due to the limited number of included articles, we are currently unable to do subgroup analysis based on the amount of rAd-p53 and the TACE regimen, and we will continue to follow the progress of studies in this area and continue to study this topic in the future.

---

## [Decision Letter · Decision Letter 1]

16 Nov 2023

PONE-D-23-04833R1Recombinant human adenovirus p53 combined with transcatheter arterial chemoembolization for liver cancer: a meta-analysisPLOS ONE

Dear Dr. xiaojin,

Thank you for submitting your manuscript to PLOS ONE. After careful consideration, we feel that it has merit but does not fully meet PLOS ONE’s publication criteria as it currently stands. Therefore, we invite you to submit a revised version of the manuscript that addresses the points raised during the review process.

We look forward to receiving your revised manuscript.

Kind regards,

Mirosława Püsküllüoğlu, MD, PhD

Academic Editor

PLOS ONE

Journal Requirements:

**Additional Editor Comments:**

At this stage there are only few minor points we would like you to take into consideration as suggested by the reviewers.

Apologies for long review process, but finding reviewers fitting this type of manuscript is a challenging task.

Reviewers' comments:

Reviewer's Responses to Questions

**Comments to the Author**

1. If the authors have adequately addressed your comments raised in a previous round of review and you feel that this manuscript is now acceptable for publication, you may indicate that here to bypass the “Comments to the Author” section, enter your conflict of interest statement in the “Confidential to Editor” section, and submit your "Accept" recommendation.

Reviewer #1: All comments have been addressed

Reviewer #4: (No Response)

2. Is the manuscript technically sound, and do the data support the conclusions?

Reviewer #1: Yes

Reviewer #4: Yes

3. Has the statistical analysis been performed appropriately and rigorously? 

Reviewer #1: Yes

Reviewer #4: Yes

4. Have the authors made all data underlying the findings in their manuscript fully available?

Reviewer #1: Yes

Reviewer #4: Yes

5. Is the manuscript presented in an intelligible fashion and written in standard English?

Reviewer #1: Yes

Reviewer #4: Yes

6. Review Comments to the Author

Reviewer #1: The revised version of the paper is OK. I don't have other comments to do. I congratulate the authors for their important work. Thank you!

Reviewer #4: Nicely designed study and important clinical question, I have few queries and recommendations:

1- What do you mean by primary liver tumor? HCC? Cholangio ?? both??

2- What is the authors explanation of higher GI symptoms and myelosuppression in TACE only group ?

3- Discussion should be broken into smaller paragraphs. Please indent the beginning of each paragraph and justify your whole text.

7. PLOS authors have the option to publish the peer review history of their article (what does this mean?). If published, this will include your full peer review and any attached files.

Reviewer #1: No

Reviewer #4: No

---

## [Author Response · Author response to Decision Letter 1]

18 Nov 2023

Dear Reviewers ：

Thank you for your letter and the reviewers’ comments concerning our manuscript entitled “

Recombinant human adenovirus p53 combined with transcatheter arterial chemoembolization for liver cancer: a meta-analysis” (PONE-D-23-04833). We really appreciate editors and reviewers greatly for their positive and constructive comments and suggestions on our manuscript. Those comments and suggestions are all valuable and very helpful for revising and improving our paper. We have read through comments carefully and have made corrections. Based on the instructions provided in your letter, we uploaded the file of the revised manuscript. Revisions in the text are shown using red highlight for additions.We would love to thank you for allowing us to resubmit a revised copy of the manuscript and the time and effort you put into our manuscripts.

Answer to the question of reviewer four

Question 1：What do you mean by primary liver tumor? HCC? Cholangio ?? both??

My answer：

Thank you very much for your question and I apologize for not stating that it was our problem. Primary liver cancer (PLC) is a malignant tumor that occurs in the epithelial cells of hepatocytes or intrahepatic bile ducts. Primary liver cancer mainly includes three pathological types: hepatocellular carcinoma (HCC), intrahepatic cholangiocarcinoma (ICC), and combined hepatocellular-cholangiocarcinoma (cHCC-CCA). Among them, hepatocellular carcinoma (HCC) accounts for about 85-90% of all primary liver malignancies. 

Cholangiocarcinoma (CCA) is a malignant tumor originating from the extrahepatic bile duct.According to their location, they are classified as carcinoma of the perihilar bile ducts,carcinoma of the hilar bile ducts and carcinoma of distal extrahepatic bile ducts.

We re-examined the 17 studies we included and found that 11 papers included patients with primary liver cancer and another six included patients with hepatocellular hepatocellular carcinoma, and given that hepatocellular hepatocellular carcinoma makes up a major portion of primary liver cancers, we believe that the inclusion of patients with hepatocellular hepatocellular carcinoma was almost exclusively in this meta-analysis.

Question 2: What is the authors explanation of higher GI symptoms and myelosuppression in TACE only group ?

My answer：

We investigated the cause of side effects in the two groups because we wanted to prove that the addition of rAd-p53 in the experimental group did not increase the original side effects of TACE, and to prove the safety of Rad-p53. the results of meta-analysis proved that the incidence of side effects in the experimental group was lower than that in the control group. Given that only Rad-p53 was different between the experimental group and the control group, we believe that Rad-p53 may have altered the immune response of the patients in the experimental group, leading to a reduction in side effects, which is also a point of great interest to us, and we will further study the alteration of the immune environment of the patients by Rad-p53 in the future. In addition, given that the patients in the experimental group all signed the informed consent for Rad-p53, we think that there may be some psychological implication for the patients in the experimental group.

Question 3：Discussion should be broken into smaller paragraphs. Please indent the beginning of each paragraph and justify your whole text.

My answer：

Thank you very much for the heads up, sorry for the error, we have revised our discussion section as you requested.

---

## [Editor Report · Decision Letter 2]

21 Nov 2023

Recombinant human adenovirus p53 combined with transcatheter arterial chemoembolization for liver cancer: a meta-analysis

PONE-D-23-04833R2

Dear Dr. Xaojin,

We’re pleased to inform you that your manuscript has been judged scientifically suitable for publication and will be formally accepted for publication once it meets all outstanding technical requirements.

Kind regards,

Mirosława Püsküllüoğlu, MD, PhD

Academic Editor

PLOS ONE

---

## [Editor Report · Acceptance letter]

11 Dec 2023

PONE-D-23-04833R2 

Recombinant human adenovirus p53 combined with transcatheter arterial chemoembolization for liver cancer: a meta-analysis 

Dear Dr. Wu:

I'm pleased to inform you that your manuscript has been deemed suitable for publication in PLOS ONE. Congratulations! Your manuscript is now with our production department. 

Kind regards, 

on behalf of

Dr. Mirosława Püsküllüoğlu 

Academic Editor

PLOS ONE